# communications
# earth & environment

# Joint optimization of land carbon uptake and albedo can help achieve moderate instantaneous and long-term cooling effects

Alexander Graf [1✉], Georg Wohlfahrt [2], Sergio Aranda-Barranco [3,4], Nicola Arriga [5], Christian Brümmer [6], Eric Ceschia [7], Philippe Ciais [8], Ankur R. Desai [9], Sara Di Lonardo [10], Mana Gharun [11], Thomas Grünwald [12], Lukas Hörtnagl [13], Kuno Kasak [14], Anne Klosterhalfen [15], Alexander Knohl [15], Natalia Kowalska [16], Michael Leuchner [17], Anders Lindroth [18], Matthias Mauder [12], Mirco Migliavacca [5], Alexandra C. Morel [19], Andreas Pfennig [20], Hendrik Poorter [21,22], Christian Poppe Terán [1], Oliver Reitz [17], Corinna Rebmann [23], Arturo Sanchez-Azofeifa [24], Marius Schmidt [1], Ladislav Šigut [16], Enrico Tomelleri [25], Ke Yu [8], Andrej Varlagin [26] & Harry Vereecken [1]

Both carbon dioxide uptake and albedo of the land surface affect global climate. However, climate change mitigation by increasing carbon uptake can cause a warming trade-off by decreasing albedo, with most research focusing on afforestation and its interaction with snow. Here, we present carbon uptake and albedo observations from 176 globally distributed flux stations. We demonstrate a gradual decline in maximum achievable annual albedo as carbon uptake increases, even within subgroups of non-forest and snow-free ecosystems. Based on a paired-site permutation approach, we quantify the likely impact of land use on carbon uptake and albedo. Shifting to the maximum attainable carbon uptake at each site would likely cause moderate net global warming for the first approximately 20 years, followed by a strong cooling effect. A balanced policy co-optimizing carbon uptake and albedo is possible that avoids warming on any timescale, but results in a weaker long-term cooling effect.

[1] Institute of Bio- and Geosciences: Agrosphere (IBG-3), Research Centre Jülich, Jülich, Germany. [2] Universität Innsbruck, Institut für Ökologie, Innsbruck, Austria. [3] Andalusian Institute for Earth System Research (IISTA-CEAMA), 18071 Granada, Spain. [4] Departament of Ecology, University of Granada, 18071 Granada, Spain. [5] European Commission, Joint Research Centre (JRC), Ispra, Italy. [6] Thünen Institute of Climate-Smart Agriculture, Braunschweig, Germany. [7] CESBIO, Université de Toulouse, CNES/CNRS/INRA/IRD/UPS, Toulouse, France. [8] Laboratoire des Sciences du Climat et de l'Environnement, LSCE/IPSL, CEA-CNRS-UVSQ, Université Paris-Saclay, Gif-sur-Yvette 91191, France. [9] Department of Atmospheric and Oceanic Sciences, University of Wisconsin-Madison, Madison, WI, USA. [10] Research Institute on Terrestrial Ecosystems-National Research Council (IRET-CNR), Sesto Fiorentino, Italy. [11] Institute of Landscape Ecology, University of Münster, Münster, Germany. [12] Technische Universität Dresden, Institute of Hydrology and Meteorology, Dresden, Germany. [13] Department of Environmental Systems Science, ETH Zürich, Universitätstraße 2, Zürich 8092, Switzerland. [14] Department of Geography, University of Tartu, Tartu, Estonia. [15] Bioclimatology, University of Göttingen, Göttingen, Germany. [16] Global Change Research Institute CAS, Bělidla 986/4a, CZ-60300 Brno, Czech Republic. [17] Physical Geography and Climatology, Institute of Geography, RWTH Aachen University, Aachen, Germany. [18] Department of Physical Geography and Ecosystem Science, Lund University, Lund, Sweden. [19] Division of Energy, Environment and Society, University of Dundee, Dundee, UK. [20] Department of Chemical Engineering, University of Liège, Liège, Belgium. [21] Institute of Bio- and Geosciences: Plant Sciences (IBG-2), Research Centre Jülich, Jülich, Germany. [22] Department of Natural Sciences, Macquarie University, North Ryde, NSW 2109, Australia. [23] Department Computational Hydrosystems, Helmholtz Centre for Environmental Research (UFZ), Permoserstr. 15, 04318 Leipzig, Germany. [24] Earth and Atmospheric Sciences Department, Centre for Earth Observation Sciences (CEOS), Edmonton, AB, Canada. [25] Faculty of Agricultural, Environmental and Food Sciences, Free University of Bolzano, Piazza Università 5, 39100 Bolzano, Italy. [26] A.N. Severtsov Institute of Ecology and Evolution, Russian Academy of Sciences, 119071Leninsky pr.33, Moscow, Russia. ✉email: a.graf@fz-juelich.de

The world's land surfaces affect global climate biochemically via the release and uptake of atmospheric constituents, and biophysically through the surface energy and momentum budget[1,2]. The $CO_2$ sink strength (net ecosystem productivity, NEP) is the most discussed biogeochemical, and surface albedo ($\alpha_s$) the most discussed biophysical property of terrestrial ecosystems. Greenhouse-gas and albedo-based effects from land management can be compared either via radiative-forcing type, or $CO_2$ equivalent metrics[3]. However, both effects are not equivalent[4]. One important difference is the temporal scale on which they affect climate: a change in albedo causes an immediate corresponding change in the radiative balance, while a change in NEP causes a continuous atmospheric $CO_2$ accumulation or depletion, and thus a cumulative, lagged alteration in radiative forcing[5]. Typically, afforestation leads to higher NEP, but lower albedo[6–9]. This is particularly true in boreal biomes, largely because evergreen forests mask the high albedo of snow-covered ground[10].

Such trade-offs cause challenges for global warming mitigation policies[1–3,9,11,12]. Global emissions from fossil-fuel burning are still increasing, and by the mid-21$^{st}$ century greenhouse-gas concentrations are likely higher than recommended to safely avoid tipping points[13]. Keeping temperatures as low as possible during this period could be facilitated by a strong cooling effect of the land surface, to which short-term albedo effects can make a considerable contribution[14]. On the other hand, enhanced terrestrial $CO_2$ uptake and associated long-term carbon storage is part of the plan to reduce global atmospheric $CO_2$ concentrations in the second half of the century, and thus avoid long-term damage[15]. Therefore, it is important to quantify and understand the relation between albedo and $CO_2$ uptake of land ecosystems. Ideally, a change to climate-conserving land management practices would increase both, and thus provide a cooling effect on all time scales[6,16,17]. Apart from forest and snow coverage[10,18,19], little has been published about possible systematic relations

between NEP and $\alpha_s$. Smith, et al.[20] suggested a reduced albedo as a result of replacing non-forest vegetation by taller non-forest vegetation, e.g. during conversion of grass or traditional crops to energy crops, and Genesio, et al.[21] a negative albedo effect of biochar application. In contrast to the general tendency, some authors demonstrated cases where land-surface change could have a cooling effect in terms of both albedo and $CO_2$ uptake[22–24], including cover crop application in agriculture[25–29].

Here we analyzed the co-variability between NEP and $\alpha_s$ across a large range of forested and forest-free, snow-affected, and snow-free land surfaces from a network of direct albedo and $CO_2$ flux measurements. Our data suggest a global relationship between the maximum attainable NEP and $\alpha_s$ of land ecosystems beyond the known effects of forests and snow-masking. Using different hypothetical mitigation strategies acting on carbon fluxes and $\alpha_s$, we evaluated the resulting temporal pathways of top-of-atmosphere net radiation changes. Because many sites currently exhibit an NEP- $\alpha_s$ combination below the maximum relationship, a balanced land use and management change increasing both appears to be possible. Finally, we discuss explanations for the shape of this co-variability and open future questions, particularly with regard to land-management optimization.

## Results and discussion

**Global relation between productivity and albedo.** The relation between (multi-)annual NEP and $\alpha_s$ across sites is inverse, but also strongly heteroscedastic. Maximum attainable values of both appear to limit each other and are bound by a roughly hyperbola-like envelope (Fig. 1a). Forests (all sites classified as forests in the land classification system of the International Geosphere-Biosphere Programme IGBP) cover an $\alpha_s$ range from 0.07 to 0.27. Non-woody sites (IGBP classes grassland, cropland and snow) cover an $\alpha_s$ range from 0.16 to 0.48. IGBP classes of transitional or unspecified plant cover (savanna, shrubland, wetland) extend over almost the whole range of aforementioned

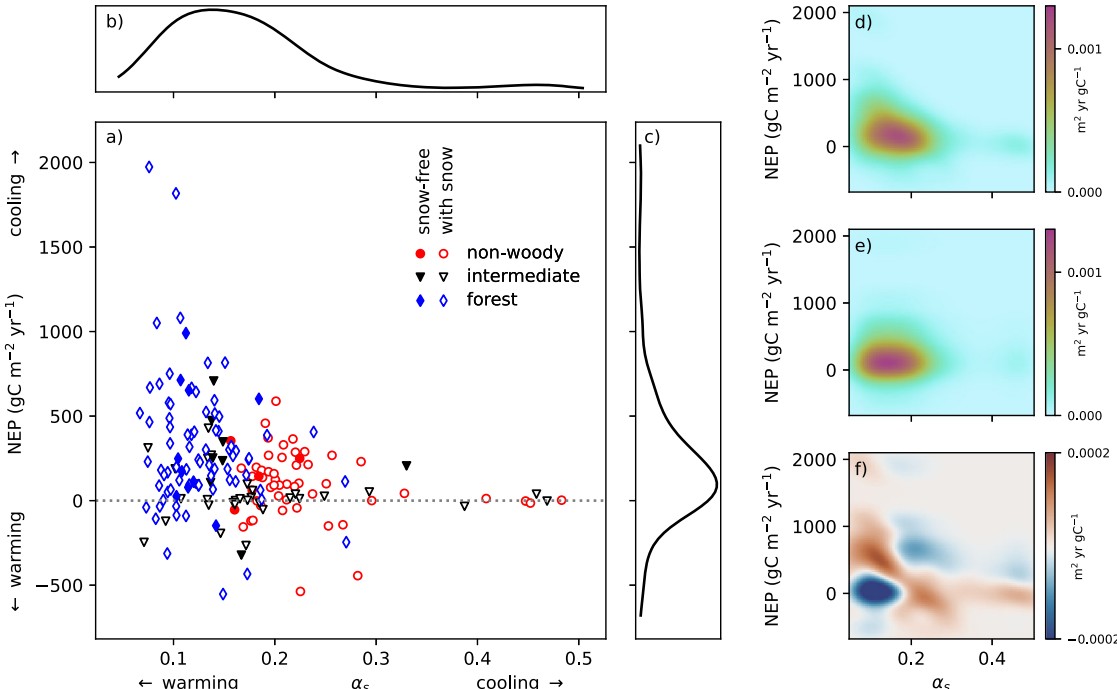

**Fig. 1 Global relation between site-averaged NEP and $\alpha_s$. a** Mean (multi-)annual net ecosystem productivity (NEP) versus surface albedo ($\alpha_s$) across all sites. All sites, where our algorithm detected snow at least occasionally, were additionally marked with a white interior. Spearman correlation coefficient is −0.30. Panels **b**, **c** show univariate kernel density estimates of the marginal distributions of $\alpha_s$ and NEP. **d** Bivariate (2-dimensional) kernel density estimate. **e** Product of the marginal distributions (expected bivariate kernel density for uncorrelated NEP and $\alpha_s$). **f** Difference of both.

albedo values. Each of these groups, even after additionally filtering for completely snow-free sites (filled points in Fig. 1a), individually appears to obey roughly the same limit observed for all sites globally. Forests and their masking effect on snow, which previous discussion focused on[10,18,19], thus cannot be the only reason for the relationship.

Figure 1b–f further demonstrates that the bivariate distribution of NEP and $\alpha_s$ (Fig. 1d) is not identical to an uncorrelated combination (e) of the univariate distributions of both variables (b and c). The range of differences between the actual and uncorrelated distribution accounts for about one third of the range of densities in b and c, and reveals that high albedo is more likely associated with low NEP, and vice versa. The region of increased density, like the apparent limit in Fig. 1a, follows a hyperbolic pattern (Fig. 1f). In general, the fact that a high albedo and large NEP are often incompatible is documented[20] and may be expected from the fact that plant covers have a lower albedo than most natural unvegetated surfaces. However, to our knowledge the global nature, beyond forest effects, of this hypothetically natural relationship has not been demonstrated, and its shape as seen in Fig. 1 has not been quantified. Our findings further suggest it applies as an envelope to maximum NEP and $\alpha_s$ values, but not to the bulk of the examined, and mostly economically used, sites. This suggests much of the land surface can, and possibly did, provide better climate services than under current management by having an NEP and $\alpha_s$ combination closer to their joint natural limit. We conceptually examine this possibility in the following section. Because of potential seasonal co-variability of $\alpha_s$ and irradiation this will be done with monthly albedo observations[30], the role of which in causing our findings will further be examined in a further subsection.

**Different scenarios for mitigating climate change**. Without claiming that land-use changes at this scale are feasible or advisable considering other sustainable development goals, we evaluated four scenarios (methods). In the first two scenarios, we hypothetically maximized each site's NEP (SC1, using the highest value occurring at a site with matching climate and that site's albedo) or albedo (SC2, vice versa). In a balanced scenario, we found the partner site with the largest joint relative improvement of NEP and $\alpha_s$ where possible, or the smallest relative trade-off otherwise (SC3). Finally, we computed the outermost limit of joint maximisation, assuming that the largest NEP and $\alpha_s$ found among climatic partner sites can be freely combined (SC4). This implies a breakthrough in breeding or finding yet unexamined ecosystems, which is likely unfeasible but illustrates an outer limit to the attainable effect at any timescale.

The resulting modelled top-of-atmosphere global net radiation change from both NEP and albedo changes over 100 years following the hypothetical land-use change is shown in the lower part of the figure. The shown uncertainty band results from assumptions on the time period required for final ecosystem NEP and albedo to establish, saturation effects as cumulative NEP approaches zero or a maximum capacity, carbon exports by harvest, and albedo kernels (Methods).

SC2 (albedo optimization at any cost) had the strongest immediate cooling effect in the first years after change. However, due to carbon loss, SC2 showed a possible net warming effect in the long term (from approximately 30 years after change onwards). Conversely, SC1 (NEP optimization) had an albedo-caused warming effect during the first years, which after about 20 years turned into the strongest cooling effect except for SC4. In SC3 (balanced), the initial warming effect could be avoided, instead a small immediate cooling effect was

created at the cost of less cooling later. SC4 (breakthrough) showed the strongest cooling effect both on short-term (together with SC2) and long-term (with a constant advantage over the next best scenario SC1, the relative magnitude of which however diminishes over time).

Further uncertainties, which we cannot quantify with currently available model-based evidence for our specific scenarios, include the atmospheric adjustments (ref. [15], therein chapter 7.3) and feedbacks (chapter 7.4) following land-use change. An inter-comparison of effective (ERF) vs. instantaneous radiative forcing (IRF) of land-use change across CMIP6 models[31] exhibited both reinforcing and offsetting adjustment effects depending on model. Their inter-model mean importance of $-36\%$ of IRF suggests a corresponding overestimation of net radiation change in Fig. 2. A reduction of this magnitude is plausible given that e.g. forests mostly stimulated low-level clouds in boreal and temperate regions, which dominate our dataset, in a recent study[32]. The dominant stimulated cloud types have net cooling effects at the TOA[33] and would thus counteract the heating effect of lower forest albedo. However, the forest breeze circulation contributing to such cloud stimulation[34] can also reduce cloudiness over nearby non-forest surfaces. Assuming that albedo strongly contributes to the land-use effects inhibited according to Smith, et al. [31], while the effect of $CO_2$ alone has been shown to be slightly reinforced by tropospheric adjustments by approximately 5% (ref. [15], therein Table 7.4), adjustments can also affect the balance between albedo and $CO_2$ effects seen in Fig. 2. A weakened albedo and reinforced $CO_2$ effect would underline our finding that pure albedo optimization (SC2) is prone to result in long-term warming, and suggest an even earlier onset of net cooling for NEP optimization (SC1).

**Evidence on causes and exceptions**. Several reasons have been, or can be invoked to causally explain a negative relationship between NEP and albedo:

*Light harvesting* Photosynthesis requires the absorption of photosynthetically active radiation (*PAR*, roughly half of shortwave incoming radiation $SW_{in}$[35]). For different stand properties and seasonal cycles in boreal forests, case studies[36,37] found weak negative linear correlations between canopy-scale reflectance and the fraction of absorbed *PAR* (fAPAR). Likewise, satellite-derived fAPAR of our study sites shows negative relationships with their respective $\alpha_s$ (Fig. 3a). However, photosynthesis is known to consume only a small fraction of incoming radiation, even when focusing on its photosynthetically active part *PAR* or its absorbed part. In our dataset, this fraction could be estimated by applying the energy intensity of photosynthesis ($0.469\,\mathrm{J}\,\mu mol^{-1}\,CO_2$[38,39]) to gross primary productivity GPP and comparing to non-reflected incoming shortwave radiation $SW_{net}$. While the resulting fraction of radiation energy used for photosynthesis can vary depending on ecosystem and cloudiness[40], its site-averaged long-term value was nowhere above 2.7% and on average (both arithmetic mean and median) 1.2% (standard deviation 0.6%). This fraction showed no clear relation to $\alpha_s$ (see Fig. 3b including correlations). The majority of absorbed *PAR* is transformed to sensible or latent heat, and high near-infrared reflectance of healthy vegetation[41] further contributes to decoupling overall albedo from absorbed *PAR*. Consequently, at least in theory, ecosystems could combine high photosynthetic productivity with a high albedo, and it remains to be examined whether the reflection of more unused *PAR* necessary for this combination is either physically impossible, was not evolutionary beneficial, or might already exist in special cases[22].

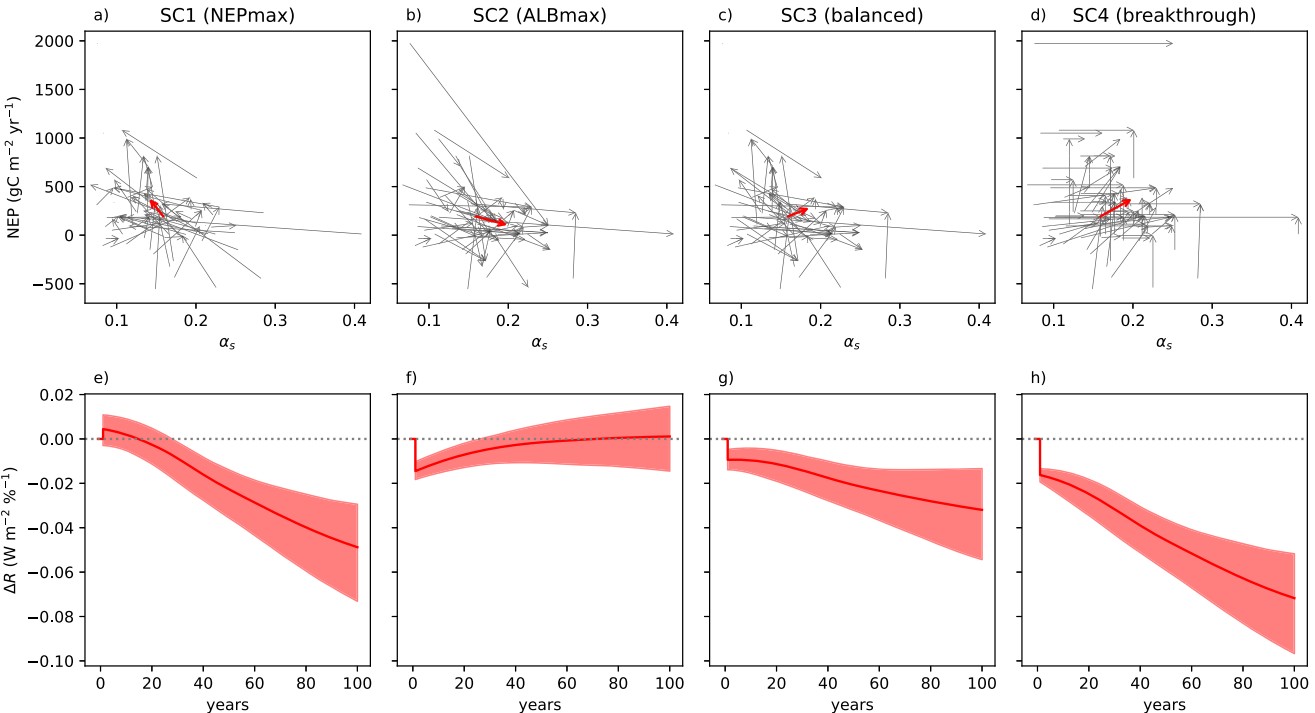

**Fig. 2 Changes in net ecosystem productivity NEP, surface albedo $\alpha_s$ and radiation for four different scenarios. a–d** Hypothetical change in $\alpha_s$ and NEP for each site (grey arrows) and averaged across all sites (red arrows). **e, f** Resulting pathways of global top-of-atmosphere net radiation change $\Delta R$ per each % of land surface on which land use is changed according to the panel above. The ensemble mean (bold red line) and full uncertainty range (shade, min to max) result from albedo kernel uncertainties between studies, transition times, C saturation effects and harvest exports (see methods). Panel **a** and **e** refer to an NEP maximization scenario, **b** and **f** to albedo maximization, **c** and **g** to a balanced scenario maximizing joint relative increase in both, and **d** and **h** a hypothetical scenario assuming NEP and albedo can be maximized independent of each other.

*Low albedo as a result of growth.* Tall and dense canopies with an efficient leaf economic spectrum can be both the cause and the result of large NEP[42]. Such canopies, because of the complex vertical structure, are more likely to support multiple reflections, effectively capturing more and reflecting less light[43]. Case studies[36,37] found negative correlations of reflectance with various biomass-density measures. Forests are an extreme case, but as suggested for energy crops[20], the causal relation as such can be expected to apply to non-woody canopies as well. In our dataset, canopy height was negatively related to albedo (Fig. 3c); significantly ($p < 0.05$) negative Spearman correlation coefficients of 0.37 to 0.83 were also found when analysing snow-affected and snow-free ecosystems, or forests, non-woody and intermediate ecosystems separately.

*Respiration.* NEP is determined by ecosystem respiration as well as by photosynthesis. Small heterotrophic respiration losses of $CO_2$ can be favoured by water excess (such as in peatlands), seasonal temperature and moisture limitation dynamics (such as on chernozems), or chemical composition of plant material and resulting litter. Components resisting decomposition, such as bark or multi-annual leaves, or humus, might generally exhibit a lower albedo than others. We are not aware of published studies on such a general relation, except for biochar application that caused decrease in surface albedo[21,44] and a weak relation between albedo and soil organic carbon found by Post, et al. [45]. Our dataset, too, allows only limited insights, since the included inferred values of ecosystem respiration $R_{eco}$ can be expected to scale with the amount of already accumulated carbon, and thus need to be normalized by it before resistance of an ecosystem or its components to respiratory losses can be detected. However, tentatively using canopy height $h_c$ as a proxy of accumulated

carbon, based on its close relation to at least aboveground biomass[46], indeed suggests the possible existence of such a relation (Fig. 3d). This is not strictly conclusive, because a negative relation between $h_c$ and albedo has already been demonstrated above.

Apart from these possible explanations for the existence of a trade-off between net $CO_2$ uptake and albedo, also some exceptions and further findings explaining the wide shape of the hyperbolic curve can be identified:

The albedo of *surfaces other than vegetation* and of photosynthetically inactive vegetation is an important component of the net albedo of sparsely or temporarily vegetated surfaces, and varies widely from values considerably higher than vegetation for snow, many dry soils poor in organic carbon and many weathered rock outcrops, to low values for water, little weathered basaltic rock and moist topsoils rich in organic carbon. Repeating the comparison shown in Fig. 1 on a monthly basis and using satellite-derived monthly fAPAR averages as a proxy for the abundance of active vegetation (see Methods) revealed that the majority of sparsely vegetated surfaces, as identified by low fAPAR, had considerably higher albedo than biologically active ecosystems that either absorb or sometimes release large amounts of $CO_2$ (Fig. 4a). For dormant or senescent plants, the decomposition of pigments (including but not limited to Chlorophyll) may contribute to this finding. One important exception is the exposition of dark soil surfaces during fallow periods in midlatitude, humid crop systems, which negatively affects both NEP and $\alpha_s$ at the same time. The introduction of cover crops would considerably increase the albedo of the European land surface[25,26]. However, cover crops alter the soil water budget[47,48], requiring a prior evaluation of their applicability and its optimal timing especially at sites with seasonal soil-

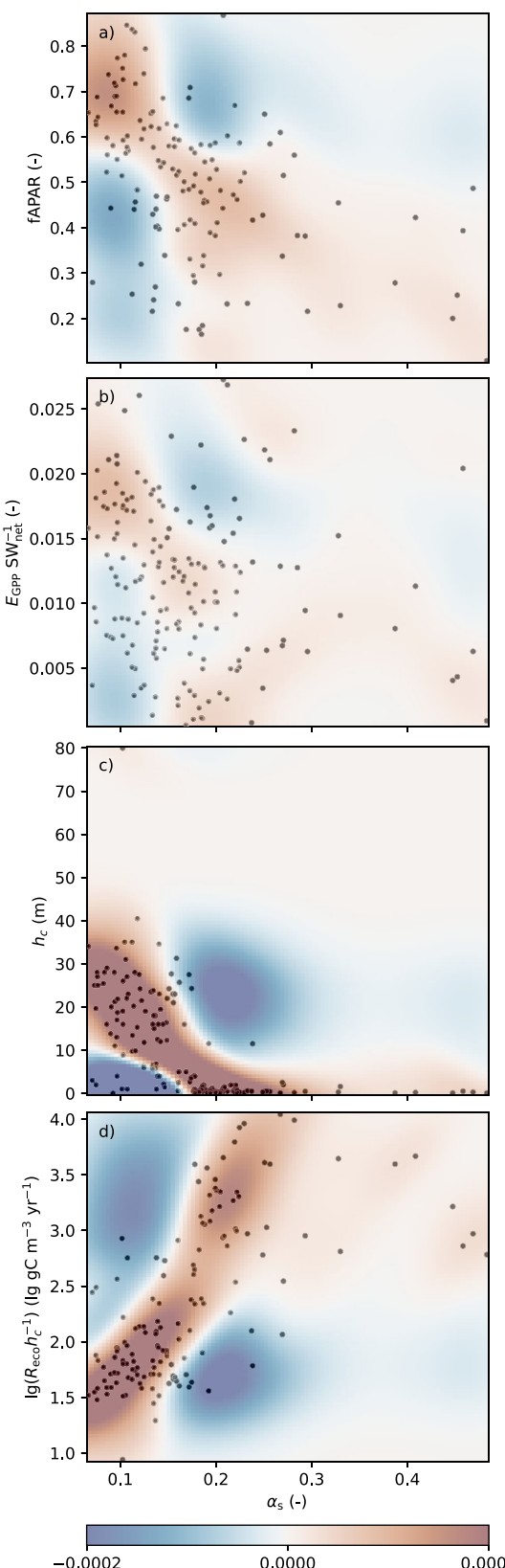

**Fig. 3 Co-variability of ecosystem properties with surface albedo $\alpha_s$, averaged per site. a** fraction of absorbed photosynthetically active radiation fAPAR, **b** fraction of non-reflected incoming radiation $SW_{net}$ used for photosynthesis $E_{GPP}$, **c** canopy height $h_c$, and **d** canopy height scaled ecosystem respiration $R_{eco}$ versus $\alpha_s$ across all sites. Background shadings show the difference between actual bivariate kernel density and expectations from univariate distributions for unrelated variables as in Fig. 1f (units of colour map: inverse of the units of the y axis of each subpanel). Spearman correlations for panels **a–d** are −0.48, −0.23, −0.73 and 0.66, respectively (all significant at $p = 0.05$). Further scatterplots with potential covariates are given in Supplementary Fig. 6.

strategy. Different studies on soybeans found their productivity and $CO_2$ uptake either to be lower[51], remain unharmed or even increase[22] compared to non-deficient plants.

It has been further shown that *leaf nitrogen content* not only favours $CO_2$ uptake, but may also influence leaf reflectance. One study[23] found tight, linear, positive relationships. Others demonstrated invariance of albedo under increasing leaf nitrogen content[52], with canopy structure likely being a crucial factor in the interplay between canopy-scale $CO_2$ uptake, leaf nitrogen content and reflectance, or presented evidence of the original finding being a possible artifact[53].

Another confounding factor is the variability of *albedo within photosynthetically strongly active ecosystems*. Snow-free albedos of deciduous forests were shown to be highest in early summer when foliage is fresh[54], and tropical forest leaf albedo to decrease with climate change due to a reduction in leaf mass area with warmer temperatures, resulting in reduced near-infrared albedo[55]. Two distinct peaks of high NEP can be identified in Fig. 4 around $\alpha_s$ of 0.08 and 0.17, respectively. It is interesting to note that a remote-sensing based study on a single Chinese catchment[56] also found a bimodal $\alpha_s$ histogram, with peaks around 0.13 and 0.16, respectively. A separate analysis of these monthly values by IGBP class (Fig. 5) suggests that the lower peak corresponds to evergreen needleleaf forests. Evergreen broadleaf forest shows the next lowest albedo peak around 0.12, suggesting a generally lower albedo of leaves or needles with a longer lifetime as compared to deciduous broad leaves[57]. A part of the limitations witnessed on an annual basis are due to the fact that the largest joint seasonal values of NEP and $\alpha_s$ cannot be, or are not, maintained over a full year (Fig. 4b). This is particularly true for the peak NEP associated with comparatively high $\alpha_s$ of cropland and deciduous broadleaf forest (Figs. 4b and 5a, c).

Focusing on values for the growing season (Table 1), deciduous broadleaf forests showed the highest albedos among forest types[18], and evergreen needleleaf forests the lowest. As expected (Introduction[10,19]), the albedo increase during snow periods ($\Delta\alpha_{snow}$) is smaller for forests and closed shrubland than for short canopies. The snow-free dormant season albedo differs only little from the growing season albedo ($\Delta\alpha_{dorm}$), but confirms the mitigation potential in cropland of avoiding fallow periods under potential growth conditions by cover crops as discussed above. When comparing the mixed forest type to all other forest types, or mixed species forests to monoculture-like forests, albedo and NEP values are competitive but not outstanding. In the past, forests composed of more than one tree species or even plant functional type have been demonstrated to be more resilient and potentially more productive[58–60].

## Conclusions

We found that the conflict between high albedo values and large carbon uptake of the land surface is not only due to the previously discussed presence or absence of forests or snow. Their possible

water stress[49,50]. In addition to cover crop application, two more possible exceptions to the general trade-off between NEP and albedo were proposed:

*Chlorophyll deficiency.* The breeding of chlorophyll-deficient plants has been suggested as an albedo-based mitigation

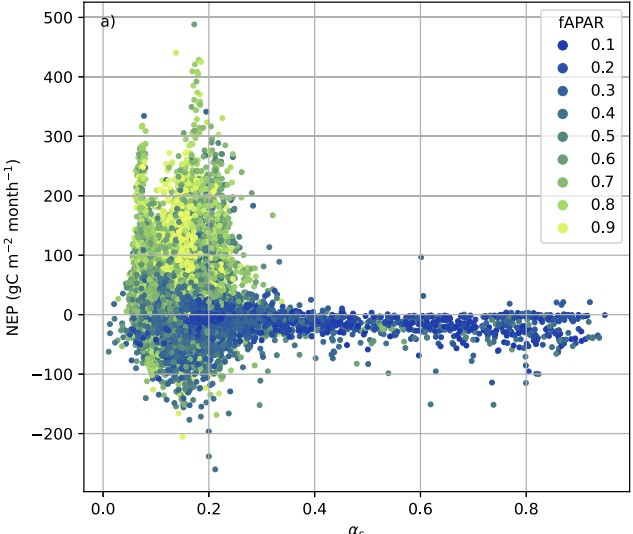
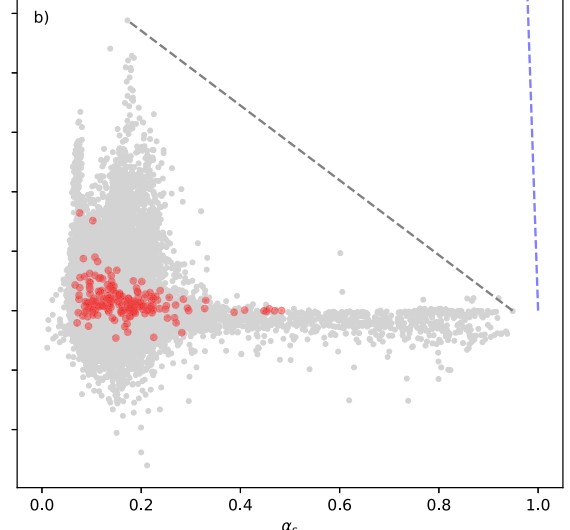

**Fig. 4 NEP vs. albedo for monthly values. a** With colours indicating monthly mean fAPAR from MODIS (MCD15A3H v6.1); **b** compared to the (multi-) annual values from Fig. 1 (rescaled to units of gC m$^{-2}$ month$^{-1}$). Blue broken line: Theoretical maximum if all absorbed PAR energy (of an assumed 100 W m$^{-2}$) was used for $CO_2$ sequestration; Grey broken line: Theoretical maximum to annual effective values if monthly averages could be freely combined under constant $SW_{in}$.

maximum values limit each other across the whole range of investigated ecosystems, and the same limitations apply when filtering for forest-free and snow-free sites. We hypothesise that this is the result of a multitude of processes, including a general tendency of conditions favouring large effective radiation absorption to promote photosynthesis, and possibly also a tendency of stable carbon stocks such as humus-rich soil or long-lived leaves to have a lower albedo. As a result, our scenario analysis confirmed that $CO_2$-based and albedo-based mitigation of global warming are generally in conflict. Without a substantial breakthrough in breeding or identifying high-productivity, high-albedo plant species (scenario 4), all scenarios include at least some trade-offs between short-term and long-term cooling effects. Maximizing albedo (scenario 2) would create a future carbon debt: its application to cause a strong immediate cooling effect now would lead to a deadlock few decades later, when not only the cooling effect possibly reverts, but any effort to correct towards a more NEP-oriented policy would add an additional instantaneous albedo-based warming. Immediate measures to maximize NEP (scenario 1) in combination with ambitious emission reductions, in contrast, is compatible with the requirement to avoid a temporary overshoot of the Paris target later in this century[13], to avoid overpassing of tipping points. It would accelerate warming in the next decade but counteract it later, as a net result advancing and at the same time lowering global peak temperatures. However, associated land-use changes such as afforestation may reduce the availability of other ecosystem resources such as water or nutrients depending on location[61–63], or could be counteracted by drought-induced insect infestations and fires[64,65], and require time. Measures in the framework of scenario 1, where possible and not in conflict with other criteria of sustainable land-use, should therefore be taken immediately. Implementing them in future decades, when approaching peak global temperature 2 °C above pre-industrial levels, increases the risk to trigger tipping points. As an alternative, a balanced mitigation scenario aiming at a joint global increase in NEP and albedo is possible, due to our finding that most of the land surface is currently not in a state close to the joint NEP-albedo-limit. As an example, productive cropland typically has a lower albedo and negative NEP during fallow periods, which can be avoided in

many cases e.g. by cover crops. A global scenario restricted to co-maximization of albedo and NEP (scenario 3) avoids warming at any time, but at the cost of being less efficient than pure NEP maximization on the long term.

Our summary of existing evidence underlines the need for more detailed studies that integrate the albedo, greenhouse-gas, and further climate effects of land use[18]. In our opinion two currently underexamined research questions require future attention. Firstly, how do albedo and carbon uptake interact during slow land-use change, particularly during afforestation measures? For example, high albedo herbaceous canopies under freshly planted or spontaneously growing trees can help to limit the initial warming effect of afforestation[7]; the effective albedo of the whole ecosystem will only increase at the same pace at which carbon was already successfully sequestered. Secondly, how do surface roughness, evapotranspiration[66] and non-$CO_2$ gases alter climate effects of land use[18,67]? Water vapour is an important but rapidly removed greenhouse gas, efficient vehicle of vertical and horizontal latent heat transport, and prerequisite for the formation of reflective clouds and precipitation[68]. Due to the multiple direct and feedback effects the net effect of changes in evapotranspiration is particularly challenging to predict. While in model-based studies the concept of effective radiative forcing (ERF[31,69–71]) can help to overcome the issue of comparing multiple direct and indirect effects of a changed variable to each other, they remain difficult to disentangle in observations. A future larger body of land-use change scenarios for which ERF is modelled, and converging results between improved models would enhance the estimation of adjustment effects in studies not involving dedicated general circulation model runs. Finally, while better quantification of wanted and unwanted land use effects on global climate is indispensable for projections and policies, their magnitude in our scenario analysis supports earlier warnings that climate-smart land management cannot replace rapid emission reductions[2,9].

## Methods
**Dataset**. We compared joint in-situ measurements of land-atmosphere fluxes of $CO_2$ and radiation by compiling and

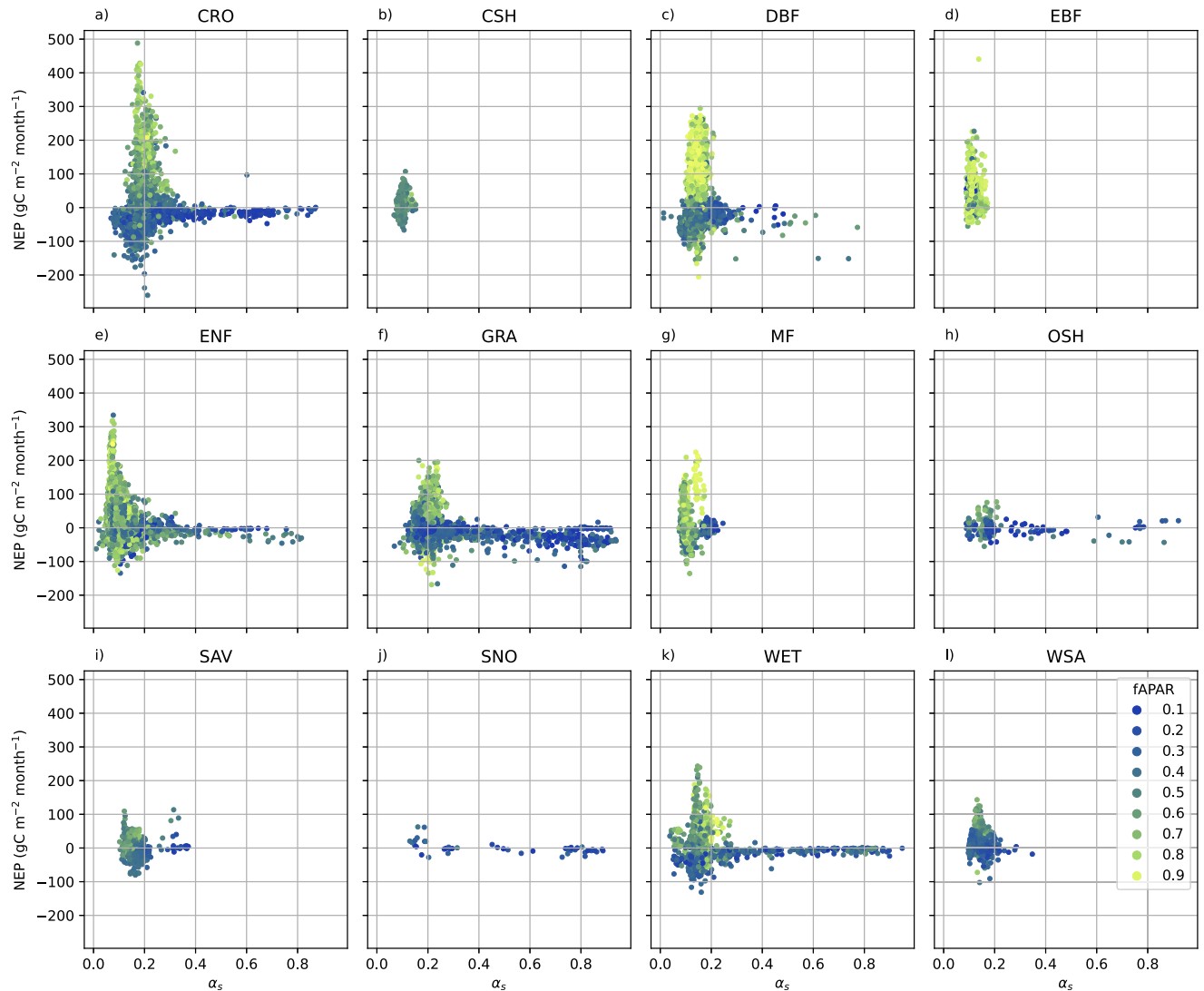

**Fig. 5 NEP vs. albedo for each IGBP ecosystem class. a** Cropland, **b** closed shrubland, **c** deciduous broadleaf forest, **d** evergreen broadleaf forest, **e** evergreen needleleaf forest, **f** grassland, **g** mixed forest, **h** open shrubland, **i** savanna, **j** snow, **k** wetland, **l** woody savanna.

postprocessing half-hourly time series from a global network of eddy-covariance stations[72]; in particular, the FLUXNET2015[73] dataset, and recent compatible updates to its European and American branches of ICOS[74] and AMERIFLUX[75], respectively. All stations with an open data policy and at least one calendar year of measurements of the core variables net ecosystem productivity (NEP based on dataset variable NEE_VUT_REF), incoming and outgoing shortwave radiation at the land surface ($SW_{in}$, $SW_{out}$), were included (Supplementary Methods 1). Precipitation data were used wherever available, but considered optional in order to not further reduce the dataset. Long-term mean annual temperature (MAT), precipitation (MAP, Supplementary Fig. 2), coordinates (Supplementary Fig. 1), average number of days with snow cover, and International Geosphere-Biosphere Programme (IGBP) ecosystem type were provided by site operators with the respective dataset and via the European Fluxes Database Cluster (http://www.europe-fluxdata.eu/home/sites-list). Where MAT or MAP were missing, the mean annual temperature and precipitation of the time series were used.

The dataset comprises 176 sites with 1 to 25 valid years each, totalling to 1167 site-years. The dataset covers 12 out of 17 IGBP land-cover classes, missing only the classes urban, mosaic, barren, water body and deciduous needleleaf, and covers the global biome

variability in temperature-precipitation space (Supplementary Material 1.1).

Imputation values for data gaps in $SW_{in}$ and NEP, and its partitioning into gross primary production (GPP) and ecosystem respiration ($R_{eco}$), were provided along with the above-mentioned data product, with methodology as described by Pastorello, et al.[73]. The dataset does not include imputation values for data gaps in $SW_{out}$. A bias in the resulting albedo $\alpha_s$ was avoided by applying an adaptive window imputation method screening the joint time series of $SW_{in}$ and $SW_{out}$ for time periods of approximately constant $\alpha_s$ (Supplementary Methods 2,[76]). Albedo and (half-hourly) minimum temperature of these periods were also used to identify the possible presence of snow or hoar frost (Supplementary Fig. 3). Only full calendar years of successfully gap-filled data were retained in order to ensure that resulting time averages of the above-mentioned variables for each station were unbiased by seasonality. For any time interval of aggregation (e.g. year), $\alpha_s$ was computed as the ratio of the interval's average $SW_{out}$ and $SW_{in}$, which is equivalent to a flux-weighted average of $\alpha_s$ and differs from a physically misleading arithmetic time average of instantaneous $\alpha_s$.

The fraction of absorbed photosynthetically active radiation (fAPAR) was taken at 500 m spatial and 4 day temporal

**Table 1 Albedo α and net ecosystem productivity NEP of land cover types and species during the snow-free growing season (NEP > 0, index act) with standard errors for those groups including more than one site, and deviations from it (Δ) during snow-free dormancy (NEP ≥ 0, index dorm) and snow or hoar-frost -affected periods (index snow, see methods for details).**

| | $\alpha_{act}$ | $NEP_{act}$ (gC m$^{-2}$ month$^{-1}$) | $\Delta\alpha_{dorm}$ | $\Delta NEP_{dorm}$ (gC m$^{-2}$ month$^{-1}$) | $\Delta\alpha_{snow}$ | $\Delta NEP_{snow}$ (gC m$^{-2}$ month$^{-1}$) |
|---|---|---|---|---|---|---|
| CRO (cropland) | 0.20 ± 0.004 | 116 ± 11 | −0.02 | −159 | +0.30 | −135 |
| CSH (closed shrubland) | 0.11 ± 0.004 | 29 ± 10 | −0.01 | −49 | +0.02 | −24 |
| DBF (deciduous broadleaf forest) | 0.15 ± 0.004 | 101 ± 8 | −0.02 | −144 | +0.08 | −118 |
| *Brachystegia spiciformis* | 0.13 | 12 | +0.02 | −45 | | |
| *Fagus sylvatica* | 0.15 ± 0.013 | 126 ± 14 | −0.02 | −169 | | |
| *Populus tremuloides* | 0.15 | 99 | −0.00 | −137 | +0.07 | −115 |
| *Quercus cerris* | 0.15 | 154 | −0.05 | −187 | −0.00 | −144 |
| Mix | 0.15 ± 0.006 | 105 ± 14 | −0.02 | −146 | +0.05 | −118 |
| EBF (evergreen broadleaf forest) | 0.11 ± 0.002 | 62 ± 11 | −0.01 | −56 | +0.03 | −6 |
| *Eucalyptus microcarpa* | 0.12 | 38 | | | | |
| *Eucalyptus regnans* | 0.10 | 151 | | | +0.02 | +5 |
| *Quercus ilex* | 0.12 | 32 | +0.01 | −53 | +0.03 | −19 |
| Mix | 0.11 ± 0.003 | 49 ± 14 | −0.00 | −54 | +0.03 | −3 |
| ENF (evergreen needleleaf forest) | 0.10 ± 0.004 | 53 ± 5 | +0.01 | −71 | +0.10 | −56 |
| *Picea abies* | 0.07 ± 0.003 | 70 ± 9 | +0.01 | −98 | +0.09 | −77 |
| *Picea mariana* | 0.08 ± 0.004 | 40 ± 5 | +0.01 | −55 | +0.07 | −50 |
| *Pinus halepensis* | 0.13 | 33 | −0.00 | −46 | | |
| *Pinus pinaster* | 0.10 ± 0.006 | 59 ± 4 | +0.00 | −81 | +0.02 | −44 |
| *Pinus pinea* | 0.12 | 52 | +0.01 | −87 | +0.04 | −43 |
| *Pinus sylvestris* | 0.11 ± 0.014 | 56 ± 0 | +0.02 | −75 | +0.14 | −73 |
| Mix | 0.09 ± 0.006 | 64 ± 20 | +0.00 | −72 | +0.06 | −60 |
| GRA (grassland) | 0.19 ± 0.004 | 45 ± 4 | −0.00 | −70 | +0.27 | −57 |
| MF (mixed forest) | 0.12 ± 0.013 | 67 ± 13 | −0.01 | −95 | +0.04 | −92 |
| OSH (open shrubland) | 0.16 ± 0.018 | 29 ± 7 | +0.00 | −45 | +0.24 | −35 |
| SAV (savanna) | 0.17 ± 0.021 | 23 ± 3 | +0.00 | −40 | | |
| SNO (snow) | 0.22 ± 0.043 | 23 ± 10 | −0.02 | −34 | +0.51 | −29 |
| WET (wetland) | 0.14 ± 0.011 | 45 ± 7 | −0.01 | −73 | +0.36 | −45 |
| *Carex acuta** | 0.19 | 57 | −0.01 | −87 | | |
| *Salix cinerea** | 0.17 | 64 | −0.01 | −106 | | |
| WSA (woody savanna) | 0.14 ± 0.007 | 31 ± 7 | +0.00 | −44 | +0.05 | −20 |
| *Olea europaea** | 0.18 | 20 | −0.00 | −38 | | |
| *Prosopis velutina** | 0.16 | 14 | +0.00 | −26 | +0.02 | −21 |
| Mix | 0.14 | 63 | | | | |
| All broad-leaved forests | 0.14 ± 0.004 | 89 ± 7 | −0.02 | −131 | +0.07 | −94 |
| All deciduous-leaved forests | 0.15 ± 0.004 | 101 ± 8 | −0.02 | −144 | +0.08 | −118 |
| All evergreen-leaved forests | 0.10 ± 0.003 | 55 ± 4 | +0.01 | −70 | +0.10 | −52 |
| All needle-leaved forests | 0.10 ± 0.004 | 53 ± 5 | +0.01 | −71 | +0.10 | −56 |

For all types where available plant species information indicated dominance of a single species at one or more sites, species-dominated values are contrasted to those of all mixed-species sites of the same type (*species marked with asterisk nevertheless do not cover the surface in a way allowing attribution of α to that species).

resolution from the MODIS fAPAR/LAI (MCD15A3H v6.1) product. Data were downloaded through the AppEEARS web portal[77] for the tower pixels and, after quality control (MODLAND_QC bits equal to zero) averaged to a monthly time scale (gaps were linearly interpolated if necessary). For five sites (DE-Akm, DE-Rus, FR-Tou, US-Snf, US-Wpt) a neighbour pixel had to be downloaded as the tower pixel was not processed correctly.

**Mitigation scenarios**. We distinguished four hypothetical global scenarios of maximizing NEP, $\alpha_s$, or both, and we compared their net radiative effect for the next 100 years by applying a comparison across pairs of sites, as described below. All applied scenarios had a simplified approach to frame the upper limit of achievable land-based mitigation, neglecting limitations caused by competition with other ecosystem services such as food, material production, or nature conservation. We assumed that changes to land use, fertilization and irrigation can change NEP and $\alpha_s$ at a given site to the values of any other site with similar mean annual temperature (±1.5 °C), shortwave incoming radiation, mean annual precipitation, and number of snow-affected days per year

(±20%). Including precipitation as a criterion for potential partner sites, but not its seasonal cycle, implies that scenarios can include irrigation from local sources such as groundwater or reservoirs, but not irrigation originating from either imported water or unsustainable groundwater drawdown. Out of the potential partner sites identified for each single site according to these criteria, we selected for:

**Scenario 1** (NEP optimization): The site with the highest (multi-) annual average NEP.

**Scenario 2** (albedo optimization regardless of NEP): The site with the highest (multi-) annual average $\alpha_s$.

**Scenario 3** (balanced): The site with the largest relative improvement parallel to the axis connecting the global minima and maxima of $\alpha_s$ and NEP, i.e.

$$x = \frac{1}{\sqrt{2}}\frac{\alpha_{new} - \alpha_{site}}{\alpha_{max} - \alpha_{min}} + \frac{1}{\sqrt{2}}\frac{NEP_{new} - NEP_{site}}{NEP_{max} - NEP_{min}}, \quad (1)$$

where subscripts max and min indicate the maximum and minimum value in the dataset shown in Fig. 1, such that the denominators normalize albedo and NEP to their total range.

Subscript site refers to the current (old) value of each site and subscript new to the value of each candidate partner site.

Scenario 4 (breakthrough): The site NEP values from scenario 1 were freely combined with $\alpha_s$ values of the partner site from scenario 2. This scenario assumed that vegetation combining a high albedo and large $CO_2$ uptake could be either bred[22] or found in places not yet represented by the global flux site network.

Out of 176 sites, only 95 had potential partners with similar climate conditions. Ensemble averages refer to this smaller subset. In some cases for a specific site and scenario combination, the site has climate partners but none that fulfil the scenario-specific improvement criterion. In this case, the site is considered to be already near the optimum land use and management state of the scenario. Consequently, while no grey arrow is shown in Fig. 2, the zero change in NEP and $\alpha_s$ still contributes to the average (red arrow). The global net radiation change at the top-of-atmosphere for each scenario, year following the hypothetical land-use change, and uncertainty ensemble member is the sum of its NEP-related radiative forcing and surface albedo effect (following subsections). The total global effect was computed assuming that the mean of the radiative change caused at all sites happens on 1% of the global land surface.

**NEP-related radiative forcings in scenarios**. The radiative forcing related to changes in NEP was computed according to Myhre, et al. [78] (Table 3 therein) and Ney, et al. [7] (Eqn. 5 and 6 therein), with updated values for average atmospheric $CO_2$ concentration (420 ppm) and airborne fraction (0.44[15]) (Supplementary Methods 4). However, changes towards larger NEP and their associated $\alpha_s$ do not occur instantly, positive or negative NEP of aged ecosystems approach zero as accumulated carbon converges to a maximum value or zero, and harvest losses not captured by NEP further affect the $CO_2$ budget of managed agricultural and forest sites. The resulting systematic uncertainty was treated by running ensemble members with different literature-based assumptions on these parameters (Supplementary Methods 5). The minimum and maximum resulting difference for each scenario is shown in Fig. 2; the time series of each ensemble member and details on the derivation of all assumptions from literature are given in the Supplementary Figure 5. In short, we considered a transition time towards higher-NEP systems of 0 to 30 years[79], maximum achievable C stocks of $50*10^3$ to $100*10^3$ gC $m^{-2}$ [80,81] after initialization at $14.498*10^3$ gC $m^{-2}$ [82], possible harvest losses of 247 up to 335 gC $m^{-2}$ $yr^{-1}$ for crops[83,84], 0 to 100 % of NEP for grassland[85], and 0 to 61 gC $m^{-2}$ $yr^{-1}$ for forests[86] and all other sites. Tropospheric adjustments can further reinforce the effective radiative forcing of $CO_2$ by approximately 5% ([15], Table 7.4 therein). This effect is not included in our computations, since current availability of model-based evidence would not allow us to consistently apply the same type of correction to albedo effects (following subsection).

**Surface albedo effect at the top of the atmosphere in scenarios**. The modification of net shortwave radiation in response to an albedo change at the same level (e.g. the land surface or top-of-atmosphere TOA) under unchanged incoming radiation is given by $SW_{in}\Delta\alpha$ at this level[14,24]. In our case, $SW_{in}$ and $\Delta\alpha$ are measured at the land surface, while effects at the TOA are of interest for comparison to $CO_2$ effects. This would require an additional correction either for cloud masking of the surface (if using $SW_{in}$ at TOA), or for atmospheric absorption (if using $SW_{in}\Delta\alpha_{in}$ at the surface). We therefore replace $SW_{in}$ by model-derived, publicly available surface albedo kernels[87].

We extracted monthly kernels for all sites from four available kernel datasets[31,87–89] and compared them, their ensemble mean,

and monthly measured $SW_{in}$ to each other (Supplementary Methods 3). All kernels were highly correlated ($R^2 \geq 0.92$) to our measured $SW_{in}$ and related to it by factors between 1.03 and 0.81 (ensemble mean 0.89, Supplementary Fig. 4). We used the largest-kernel[88] and lowest-kernel[87] dataset in the following, adding a strong albedo effect and a weak albedo effect ensemble member to the ensemble of assumptions on $CO_2$-based effects described in the previous subsection. While an annual resolution is sufficient to describe the slow evolution of net radiative effect shown in Fig. 2 and the cumulative effect of $CO_2$, annual albedo changes multiplied with annual kernels can lead to systematic errors if $\Delta\alpha_s$ and $k$ co-vary over the course of a year[27,30]. We therefore use a monthly sub-loop, corresponding to the highest temporal resolution at which globally distributed albedo kernel datasets are available, to compute the TOA albedo effect for each site and scenario from $k$ and $\alpha_s$ for each month of the year. Where a site observation or kernel dataset covered multiple years, monthly climatological $k$ and $\alpha_s$ were determined by averaging across all available years. Monthly effects were integrated over the year before combining with $CO_2$-based radiative forcings from the previous subsection to yield the net TOA radiation change, $\Delta R$, for the respective site, year and ensemble member.

The final net effect of surface albedo further differs from kernel-based instantaneous shortwave effects at the TOA, because any land use change modifies surface temperature, dust emission[70], sensible and latent heat fluxes and resulting atmospheric temperature, humidity, particle and cloud profiles, which in turn affect $SW_{in}$ and longwave radiation. Quantification of the resulting effective radiative forcing (ERF[69]) would require a global circulation model run with the exact spatiotemporal pattern and magnitude of all surface property changes accompanying the albedo change for each scenario. To provide a rough estimate of the importance of the adjustments distinguishing ERF from IRF, we refer to a study comparing among others the net effect of the same land use change across 14 Coupled Model Intercomparison Project (CMIP6) models[31].

**Methods for Table 1**. For each site, the time windows from the gap-filling process (defined by a temporally stable albedo and of variable length, see Methods: Dataset) used to determine the presence of snow were further distinguished into active (NEP > 0) or dormant (NEP ≤ 0) when snow-free. Sites were grouped according to IGBP land cover type, dominating ( ≥ 75% coverage) species where available, and the two contrasting pairs needle- vs. broadleaf and deciduous vs. evergreen for forests. Species information was compiled from Flechard, et al. [90] and Musavi, et al. [91] as well as site operators, but not available for all sites. For each group and each of the conditions active, dormant and snow, NEP was arithmetically averaged while α was the ratio of summed pairwise available $SW_{out}$ and $SW_{in}$. Dormant and snow conditions were not found at all sites, and neglected if contributing less than 10% to the total period; in these cases Δ values for the particular condition were computed against the active growing season values of only the remaining sites.

## Data availability

This study is based on publicly available data from the following sources: FLUXNET2015 product[73]: https://fluxnet.org/data/fluxnet2015-dataset/. ICOS Warm Winter 2020 product (in FLUXNET2015 format)[74]: https://doi.org/10.18160/2G60-ZHAK. AMERIFLUX FLUXNET2015 compatibility product[75]: https://ameriflux.lbl.gov/introducing-the-ameriflux-fluxnet-data-product. MODIS fAPAR[77]: https://appears.earthdatacloud.nasa.gov. CAM5 radiative kernels[89]: https://zenodo.org/record/997902[92]. HadGEM2 radiative kernels[88]: https://doi.org/10.5518/406[93]. CACK 1.0 radiative kernels[87]: https://doi.org/10.6073/pasta/d77b84b11be99ed4d5376d77fe0043d8[94].

HadGEM3-GA7.1 radiative kernels[31]: https://doi.org/10.5281/zenodo.3594673[95]. These data were quality filtered and aggregated as described in the methods section. Processed data for the figures, tables and text information in the study are stored at https://doi.org/10.5281/zenodo.8172207[96].

## Code availability

Python code for this study is stored at https://doi.org/10.5281/zenodo.8172207[96].

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

## Acknowledgements

We thank Talie Musavi, Max Planck Institute for Biogeochemistry, Jena, Germany, for providing species information originally collected for Musavi, et al. [91], Manuel Acosta, CzechGlobe, Czech republic, for provision of the data of site CZ-Krp, and the ICOS research infrastructure for data provision. L.Š. and N.K. acknowledge support by the Ministry of Education, Youth and Sports of the Czech Republic within the CzeCOS program (grant number LM2018123) and SustES—Adaptation strategies for sustainable ecosystem services and food security under adverse environmental conditions (CZ.02.1.01/0.0/0.0/16019/0000797). In 2009 funding for the Australian sites was provided to the Australia Terrestrial Ecosystem Research Network (TERN) (http://www.tern.org.au) through the Australian government's National Collaborative Research Infrastructure Strategy (NCRIS), which provides support for many OzFlux sites along with other capabilities; an overview of all Australian sites is given in Beringer, et al. [97]. Data acquisition for FR-Lam were mainly funded by the Institut National des Sciences de l'Univers of the Centre National de la Recherche Scientifique (CNRS-INSU) through the ICOS and OSR SW observatories. Facilities and staff were also funded and supported by the University Toulouse III - Paul Sabatier, the CNES and 732 IRD (Institut de Recherche pour le Développement). We are grateful to CESBIO technical team for their support at the field and to Aurore BRUT for data processing. We extend special thanks to Ecole d'Ingénieur de Purpan for accommodating the measurement devices in the plot at FR-Lam. M.G. and L.H. acknowledge funding by the Swiss National Science Foundation project ICOS-CH Phase 3 20F120_198227. A.R.D. acknowledges support for US-WCr, Us-Syv, and US-Los from the U.S. Dept of Energy Ameriflux Network Management Project subaward to the ChEAS core site cluster. K.K. acknowledges funding by Estonian Research Council (grants nr PSG631 and PSG714). G.W. acknowledges funding by the Austrian Research Promotion Agency (FFG) within the frame of the AustroSIF project. A.Kl. and

A.Kn. acknowledge funding by the German Federal Ministry of Education and Research (BMBF) as part of the European Integrated Carbon Observation System (ICOS), by the Deutsche Forschungsgemeinschaft (INST 186/1118-1 FUGG) and by the Ministry of Lower-Saxony for Science and Culture (DigitalForst: Niedersächsisches Vorab (ZN 3679)). Data acquisition for ES-Cnd was supported by the projects PID2020-117825GB-C21, PID2020-117825GB-C22, B-RNM-60-UGR20, P18-RT-3629 and grant FPU19/01647 funded by MCIN/AEI/10.13039/501100011033, "ESF Investing in your future" and FEDER/Junta de Andalucía. H.V. and A.G. acknowledge support from the Terrestrial Environmental Observatories, TERENO, funded by the Helmholtz–Gemeinschaft, Germany and the Deutsche Forschungsgemeinschaft – SFB 1502/1–2022 - Projektnummer: 450058266. A.V. was supported by the Russian Science Foundation (grant no. 21-14-00209). The Scientific colour maps hawaii and vik are used in Figs. 1 and 3 to prevent visual distortion of the data and exclusion of readers with colour-vision deficiencies[98]. We thank three reviewers for suggesting important improvements to the study.

## Author contributions

Following the Contributor Roles Taxonomy (https://credit.niso.org), authors contributed in the following roles: Conceptualization: A.G. with contributions to analysis conceptualization by G.W., N.A., E.C., A.D., P.C., M.L., A.M., A.P., H.P., C.P., O.R., A.S., K.Y. and H.V. Software and Visualization: A.G. Formal Analysis: A.G., for fAPAR G.W. Data Curation and Resources (site data): S.A., C.B., S.D.L., M.G., T.G., L.H., K.K., A.Kl., A.Kn., N.K., A.L., M.Ma., M.Mi., C.R., M.S., L.S., E.T. and A.V. Writing—original draft: AG. Writing—reviewing and editing: G.W., S.A., N.A., C.B., E.C., P.C., A.D., S.D.L., M.G., T.G., L.H., K.K., A.Kl., A.Kn., N.K., M.L., A.L., M.Ma., M.Mi., A.M., A.P., H.P., C.P., O.R., C.R., A.S., M.S., L.S., E.T., K.Y., A.V., H.V.

## Funding

## Competing interests

The authors declare no competing interests.
