## [Peer Review File · Communications Earth & Environment]

Web links to the author's journal account have been redacted from the decision letters as indicated to maintain confidentiality.

Decision letter and referee reports: first round

17th Dec 22

Dear Dr Graf,

Your manuscript titled "Co-variability of Land Surface Albedo and Carbon Uptake: Opportunities and Limitations for Climate Change Mitigation" has now been seen by 3 reviewers, whose comments are appended below. You will see that they find your work of some potential interest. However, they have raised quite substantial concerns that must be addressed. In light of these comments, we cannot accept the manuscript for publication, but would be interested in considering a revised version that fully addresses these serious concerns.

We hope you will find the reviewers' comments useful as you decide how to proceed. Should additional work allow you to

- address these criticisms (that is, either to incorporate the suggestions or provide a compelling argument why the point made by the reviewer is not valid, or relevant to the editorial threshold as outlined below)

AND

- meet our editorial thresholds as outlined below,

then we would be happy to look at a substantially revised manuscript.

Specifically, if you wish to publish with us please:

- Provide compelling new insights that go beyond previous related work into the relationship between net achievable ecosystem carbon sink and surface albedo across different biomes and regions, and discuss the underlying mechanisms
- Provide a full explanation and justification for all your methods, including how you calculate mitigation effects taking into account changes in albedo and carbon fluxes

If you choose to take up this option, please either highlight all changes in the manuscript text file, or provide a list of the changes to the manuscript with your responses to the reviewers.

If the revision process takes significantly longer than three months, we will be happy to reconsider your paper at a later date, as long as nothing similar has been accepted for publication at Communications Earth & Environment or published elsewhere in the meantime.

We understand that due to the current global situation, the time required for revision may be longer than usual. We would appreciate it if you could keep us informed about an estimated timescale for resubmission, to facilitate our planning. Of course, if you are unable to estimate, we are happy to accommodate necessary extensions nevertheless.

Please use the following link to submit your revised manuscript, point-by-point response to the reviewers' comments with a list of your changes to the manuscript text (which should be in a separate document to any cover letter) and any completed checklist:

[Link Redacted]

Please do not hesitate to contact me if you have any questions or would like to discuss the required revisions further. Thank you for the opportunity to review your work.

Best regards,

Jinfeng Chang, PhD
Editorial Board Member
Communications Earth & Environment
orcid.org/0000-0003-4463-7778

EDITORIAL POLICIES AND FORMAT

If you decide to resubmit your paper, please ensure that your manuscript complies with our editorial policies and complete and upload the checklist below as a Related Manuscript file type with the revised article:

Editorial Policy Policy requirements (Download the link to your computer as a PDF.)

For your information, you can find some guidance regarding format requirements summarized on the following checklist: (<https://www.nature.com/documents/commsj-phys-style-formatting-checklist-article.pdf>) and formatting guide (<https://www.nature.com/documents/commsj-phys-style-formatting-guide-accept.pdf>).

REVIEWER COMMENTS:

Reviewer #1 (Remarks to the Author):

Major comments

The study by Graf et al. makes use of in-situ observations of surface albedo and CO₂ fluxes in effort to quantify the climate change mitigation potential of hypothetical land use/land cover changes that “optimize” each component individually or collectively.

Aside from being a poorly-written and poorly organized manuscript with little logical flow or structure, the study overlooks a large and important body of literature on the (biogeophysical) radiative forcing impacts of land use/land cover change (LULCC) on climate, which is crucial given that radiative forcing is the metric enabling comparisons between CO₂ concentration and albedo changes – and hence in the quantification of mitigation potentials. Any surface albedo change will be accompanied by a latent and sensible heat flux change resulting in atmospheric “adjustments” that moderate the instantaneous shortwave radiative forcing (iRF) from the albedo change. The net of these and the iRF yield the so-called “effective radiative forcing” (ERF) which is a better indicator of the ensuing climate change. Model intercomparison studies have shown that the atmospheric adjustments can either completely offset the albedo change-driven iRF or reinforce it further (c.f., Table 7 in Smith et al. 2020). Hence by focusing only on the surface albedo change and the iRF, the analysis and conclusions drawn in the

study (about LULCC's biophysical climate impact) are based on a partial and incomplete accounting. Further, the method applied to calculate the albedo-change iRF is poorly described and – if I have correctly understood – highly questionable. It appears as if differences between two state variables "SW_net,TOA" are assumed analogous to an iRF. However, this definition assumes that the optical properties of the atmosphere – like absorption ("gamma") – are independent of the surface albedo – which they are not.

Given weaknesses in the methodology combined with poor organization and writing it is difficult to recommend publication in Nat. Comm. Earth & Env.

Key papers

Smith, C. J., et al. (2020), Effective radiative forcing and adjustments in CMIP6 models, Atmos. Chem. Phys. Discuss., 2020, 1-37.

Andrews, T., R. A. Betts, B. B. Booth, C. D. Jones, and G. S. Jones (2017), Effective radiative forcing from historical land use change, Climate Dynamics, 48(11), 3489-3505.

Sherwood, S. C., S. Bony, O. Boucher, C. Bretherton, P. M. Forster, J. M. Gregory, and B. Stevens (2015), Adjustments in the Forcing-Feedback Framework for Understanding Climate Change, Bulletin of the American Meteorological Society, 96(2), 217-228.

Forster, P. M., T. Richardson, A. C. Maycock, C. J. Smith, B. H. Samset, G. Myhre, T. Andrews, R. Pincus, and M. Schulz (2016), Recommendations for diagnosing effective radiative forcing from climate models for CMIP6, Journal of Geophysical Research: Atmospheres, 121(20), 12,460-412,475.

Detailed comments

Abstract

L76: Awkward sentence structure – needs rephrasing.

L77: "is discussed"? Do the authors mean "is shown"?

L78: "evidence" or "research"?

L82: "empirically achievable" is strange wording – please clarify or revise.

Main

The main article contains too many instances of poor English grammar/writing to comment in a similar to that seen below (which was carried out first).

Supporting Information material

In general, the Supporting Information is very sloppily compiled, with several overlooked formatting and grammar issues that need correcting. As one of many examples, Figure S5 appears twice back-to-back (and both with misspelled labeling). This is unacceptable for a journal like Nature Comms. Env.

L79-81: The meaning of this sentence is unclear. What is being referred to as "this term"? Clarify.

L88: Missing reference.

L103-107: Text suddenly appears in italics and the font size has been decreased.

L106-108: This detail needs more attention/elaboration given the importance of atmospheric absorption terms in the radiative transfer model underpinning the entire study analysis. It is not

possible to reproduce the absorption terms given the currently scant description, especially given the terminology “radiative forcing” used here which conventionally denotes at state perturbation. Equations A1 – A6 rely on state variables, so it is unclear how atmospheric absorption is being “estimated by comparing radiative forcings at the surface and TOA”?

Figure S5: This figure appears twice in a row, and “radiation” in the x-axis label is misspelled in both. Also, the captions differ.

Section 1.4: I get completely lost in this section. First off, what is point that is trying to be made here about the “relation between shortwave and net total radiation”? Perhaps it would be helpful to ground this text in the radiative forcing literature? The authors seem to be alluding to differences between instantaneous shortwave radiative forcings and effective radiative forcings and the importance of atmospheric adjustments, but this is not made at all clear, and leaves me guessing because – like my previous comment – the section seems to be describing physics in terms of state variables.

L147-153: I cannot follow a single methodological description here. What is “offset” and “reduced major axis”? Are the authors regressing SW vs. NETRAD, and somehow utilizing the slope and intercept terms? Why not show this figure and the robustness of this relationship? What are the “site-wise resulting factors” that are being shown in Fig. S6 – are these the slopes? And are they statistically meaningful?

Section 1.5: Either include the equation(s) for this calculation or refer the reader to the specific equations published elsewhere.

L170: What are “difference forcings”? And do the authors mean “business-as-usual” – not “business and usual”?

L175-177: This sentence is hard to follow. What are “climate-change motivated forms of land use change”?

Fig. S7: “delta RF” in y-labels – where is this defined?

Reviewer #2 (Remarks to the Author):

In this manuscript, the authors discussed the co-variability of net ecosystem productivity (NEP) and surface albedo due to land use change in the context of climate change mitigation options. First, they analyzed observations from 176 flux sites and found that NEP and albedo cannot be changed independently. Then, they estimated the time series of radiative forcing for different hypothetical land-use scenarios and found a scenario with a cooling effect in both the short- and long-term.

The manuscript is well-written and contains valuable information for considering climate change mitigation options. Therefore, I recommend the publication after minor modifications and clarifications.

Specific comments:

Fig. 2: Can you explain the unit of the y-axis of the lower panels? What does “per % land surface” mean?

L426: If there are multi-year observations, how did you estimate the initial state of the albedo?

L471: How did you evaluate the representative area of the observation sites and connect them to the global radiative forcing?

L477: I am not fully convinced by the method that calculates the top of the atmosphere net shortwave radiation by multiplying a constant by the surface net shortwave radiation. It may be necessary to explain why this method was chosen and how much uncertainty it has compared to other methods (e.g., radiative kernels as described in Pendergrass et al. (2018) cited by the authors).

Fig. S1: It may be helpful to the reader if the IGBP type for each site is shown in the figure.

Fig. S2: Please add a unit to the x-axis.

Fig. S5: There are two same figures in the word file.

Supplementary material L88: The reference is missing in my environment.

Supplementary material L213: Please add an explanation for what NEPpot means.

Reviewer #3 (Remarks to the Author):

Review, Graf et al: Co-variability of Land Surface Albedo and Carbon Uptake: Opportunities and Limitations for Climate Change Mitigation

In this manuscript, the authors study the relationships between surface albedo and CO₂ uptake across a wide set of in situ observations from a variety of station networks, primarily FLUXNET / ICOS, and AMERIFLUX. The aim is to ascertain whether or not climate change mitigation through active modulation of vegetated land cover to increase either CO₂ uptake or surface albedo (to cause climatic cooling) is likely to cause enhancing or hindering changes in the other variable. The primary answer seems to be that an expected & significant negative relationship exists between the two, and that thus far unavailable technological means would be needed to create vegetation species with both high uptake capacity and high surface albedo if cooling potential were to be maximized through this process.

The manuscript is topically appropriate for Communications Earth & Environment and the text generally flows quite well, delivering its message in a clear way. Its wider practical impact would lie in setting boundary conditions for possible future geoengineering efforts through vegetation modulation, which in itself is a useful addition to the present literature.

However, certain aspects of the calculations behind the results are not yet justified to this reviewer's satisfaction and require revision and possible reconsideration before publication may be considered. These comments are listed below, with major comments listed first.

Major comments:

1. lines 202-205: In general the statement that PAR majority is not used for photosynthesis is likely so. However, the conversion used here, based on measurements by Black et al. (1996), was based on a single Aspen forest site for a limited time. What is the robustness of the conversion and associated analysis as shown here? Also, e.g. Chen et al. (2008) state that cloudiness variability may impact photosynthetic energy-use efficiency markedly. Given 100-yr simulations presented here, has there been considerations of potential shifts in e.g. cloudiness as a source of uncertainty?
2. Methods – mitigation scenarios: Given that surface albedo depends not only on the optical properties of the surface but also on the directionality of the incoming shortwave radiation, the discussion around surface albedo seems to also omit considerations of cloudiness variability which can and will alter the balance between direct and diffuse irradiance and thus change the blue-sky albedo considered here. Given the changes made in the mitigation scenarios, particularly in #2, did the

authors also consider similarity in illumination as one of the partner site selection conditions?

3. Methods – net radiative forcing: It was not yet fully clear how the radiative forcing was calculated for the mitigation scenarios, or if the calculation is now precise enough. It seems that a static constant is used for all sites to convert radiative energy budget changes at surface to those at TOA. Given that there are published radiative kernels (from satellite data with global coverage) which yield the TOA radiative perturbation per unit change in surface albedo (for example, Bright & O'Halloran, 2019), why were these not used instead?

4. General: Also, as a cautionary note on terminology, 'radiative forcing' is currently not a favored term when discussing radiative energy balance changes due to surface albedo changes, as the driver is not seen as external to the system (see e.g. IPCC glossaries). Surface albedo feedback (SAF) is currently preferred. Please reconsider this part of the terminology.

Minor comments by line numbers:

106-107: "to violate a safe distance to tipping points" is a rather unclear expression, revise for clarity.

154-155: The negative relationship is intuitively quite clear given that the surfaces with highest natural albedos, snow/ice and desert sand, are both rather incapable of sustaining carbon-capturing vegetation. Suggest that the authors already refer to the upcoming deeper analysis on this in the section on "Evidence".

260 and elsewhere: Some inconsistent citation styles are used in the manuscript, please harmonize.

Conclusions: It is worth while to note somewhere in the text that the focus here is on (vegetated) land surfaces. Albedo modification of oceans via e.g. enhanced or artificial sea ice generation would globally fall into the second category, but its impact is not considered in the manuscript.

Citations:

Black, T. A., Den Hartog, G., Neumann, H. H., Blanken, P. D., Yang, P. C., Russell, C., ... & Novak, M. D. (1996). Annual cycles of water vapour and carbon dioxide fluxes in and above a boreal aspen forest. *Global Change Biology*, 2(3), 219-229.

Bright, R. M., & O'Halloran, T. L. (2019). Developing a monthly radiative kernel for surface albedo change from satellite climatologies of Earth's shortwave radiation budget: CACK v1. 0. *Geoscientific Model Development*, 12(9), 3975-3990.

Chen, W. J., Black, T. A., Yang, P. C., Barr, A. G., Neumann, H. H., Nesic, Z., ... & Cuenca, R. (1999). Effects of climatic variability on the annual carbon sequestration by a boreal aspen forest. *Global Change Biology*, 5(1), 41-53.

Author Responses: first round

Dear Editors and Reviewers,

please find below our point-to-point responses to all Reviewer comments.

REVIEWER COMMENTS:

Reviewer #1 (Remarks to the Author):

Major comments

The study by Graf et al. makes use of in-situ observations of surface albedo and CO₂ fluxes in effort to quantify the climate change mitigation potential of hypothetical land use/land cover changes that “optimize” each component individually or collectively.

Aside from being a poorly-written and poorly organized manuscript with little logical flow or structure, the study overlooks a large and important body of literature on the (biogeophysical) radiative forcing impacts of land use/land cover change (LULCC) on climate, which is crucial given that radiative forcing is the metric enabling comparisons between CO₂ concentration and albedo changes – and hence in the quantification of mitigation potentials. Any surface albedo change will be accompanied by a latent and sensible heat flux change resulting in atmospheric “adjustments” that moderate the instantaneous shortwave radiative forcing (iRF) from the albedo change. The net of these and the iRF yield the so-called “effective radiative forcing” (ERF) which is a better indicator of the ensuing climate change. Model intercomparison studies have shown that the atmospheric adjustments can either completely offset the albedo change-driven iRF or reinforce it further (c.f., Table 7 in Smith et al. 2020). Hence by focusing only on the surface albedo change and the iRF, the analysis and conclusions drawn in the study (about LULCC’s biophysical climate impact) are based on a partial and incomplete accounting. Further, the method applied to calculate the albedo-change iRF is poorly described and – if I have correctly understood – highly questionable. It appears as if differences between two state variables “SW_{net,TOA}” are assumed analogous to an iRF. However, this definition assumes that the optical properties of the atmosphere – like absorption (“gamma”) – are independent of the surface albedo – which they are not.

Being an observation-based study, that can make use of GCMs only where their output is made publicly available, we cannot completely accommodate all these concerns. Given the still large uncertainty between models (which also became evident in an intercomparison during revision of our methods) we are convinced that such concerns should not lead to a potentially biased body of literature missing out observation-based studies. However, we did our best to accommodate your concerns as far as possible. In particular,

- *The TOA effect of albedo changes is no more based on our own SW_{in} measurements (with a reduction factor that is apparently difficult to defend). Instead we now use published radiative kernels for the impact of surface albedo perturbations on TOA radiation for each site location and month of the year (also see reply to comment on supplement L106-108 and 1.4).*
- *ERF and all suggested literature on it is now explicitly acknowledged in the main article (which individual components of ERF already were in the original submission, e.g. latent heat flux already at the end of the conclusion section and long-wave radiation side effects in the supplement). The revised supplement contains an effort to relate ERF and atmospheric*

adjustments to the terms in our revised framework of relating surface albedo changes to TOA net effects.

Neither a direct comparison of the kernels to measured SW_{in} at the surface, nor the small impact the revised methodology had on our main results, support the hypothesis that the old methodology led to considerable errors, but we are thankful that testing both approaches gave us opportunity to increase trust in our results, and we keep the new methodology that is more in line with your comment, also because it is easier to explain based on existing literature. Similarly, we do not agree that Table 7 in Smith et al. 2020 demonstrates a large discrepancy between albedo change based IRFs and ERFs, but found it a valuable source and included it among the studies used in our revised kernel-based methodology.

Given weaknesses in the methodology combined with poor organization and writing it is difficult to recommend publication in Nat. Comm. Earth & Env.

Key papers

Smith, C. J., et al. (2020), Effective radiative forcing and adjustments in CMIP6 models, Atmos. Chem. Phys. Discuss., 2020, 1-37.

Andrews, T., R. A. Betts, B. B. Booth, C. D. Jones, and G. S. Jones (2017), Effective radiative forcing from historical land use change, Climate Dynamics, 48(11), 3489-3505.

Sherwood, S. C., S. Bony, O. Boucher, C. Bretherton, P. M. Forster, J. M. Gregory, and B. Stevens (2015), Adjustments in the Forcing-Feedback Framework for Understanding Climate Change, Bulletin of the American Meteorological Society, 96(2), 217-228.

Forster, P. M., T. Richardson, A. C. Maycock, C. J. Smith, B. H. Samset, G. Myhre, T. Andrews, R. Pincus, and M. Schulz (2016), Recommendations for diagnosing effective radiative forcing from climate models for CMIP6, Journal of Geophysical Research: Atmospheres, 121(20), 12,460-412,475.

Detailed comments

Abstract

L76: Awkward sentence structure – needs rephrasing.

Changed to: “Both, CO₂ uptake (NEP) and albedo of the land surface affect global climate.” (L77)

L77: “is discussed”? Do the authors mean “is shown”?

Was referring to existing knowledge, which “is discussed” and “is shown” would both leave unclear. Sentence changed to “Climate change mitigation by increasing NEP however can cause a trade-off by decreasing albedo, with most research focusing...” (L77-79)

L78: “evidence” or “research”?

“research” indeed fits better here, thank you, replaced (L79).

L82: “empirically achievable” is strange wording – please clarify or revise.

Changed to “Based on the possible NEP-albedo combinations we empirically found” (L83).

Main

The main article contains too many instances of poor English grammar/writing to comment in a similar to that seen below (which was carried out first).

We scanned the whole article once more for language issues and improved it wherever deemed necessary. However, in agreement with two other Reviewers explicitly providing positive comments on language and clarity and the editor, we could not find a large amount of such instances. Note that large parts of the methods section are new now, due to the revised methodology and clarity comments of yourself and the editor.

Supporting Information material

In general, the Supporting Information is very sloppily compiled, with several overlooked formatting and grammar issues that need correcting. As one of many examples, Figure S5 appears twice back-to-back (and both with misspelled labeling). This is unacceptable for a journal like Nature Comms. Env.

We apologize for the two layout errors. See below for details.

L79-81: The meaning of this sentence is unclear. What is being referred to as “this term”? Clarify.

“this term” could have been more clearly and precise “the effect of any cloud cover that remains identical between both scenarios”. However, the whole part became obsolete by revising our methodology to better accommodate your main concern, follow comment 3 of Reviewer 3, and comment on L477 of Reviewer 2. We do not make the above assumption any more.

L88: Missing reference.

Was a reference to the next figure, now obsolete because the revised kernel-based methodology (see above) changed the whole section 1.3.

L103-107: Text suddenly appears in italics and the font size has been decreased.

Apologies. This text does not exist any more due to the method revision changing the whole section 1.3 (see above).

L106-108: This detail needs more attention/elaboration given the importance of atmospheric absorption terms in the radiative transfer model underpinning the entire study analysis. It is not possible to reproduce the absorption terms given the currently scant description, especially given the terminology “radiative forcing” used here which conventionally denotes at state perturbation. Equations A1 – A6 rely on state variables, so it is unclear how atmospheric absorption is being “estimated by comparing radiative forcings at the surface and TOA”?

All SW quantities in the manuscript are not state variables, but fluxes. Your comment probably corresponds to the distinction between radiation fluxes in a single scenario (which are also a radiative forcing according to the first definition of the official glossary of the American Meteorological society, cf. <https://glossary.ametsoc.org>), as opposed to radiative forcing defined as the result a perturbation / change (2nd definition of the above and focused on in e.g. IPCC AR5 Chapter 5). The reasoning underlying our old methodology was that under the assumption of an unchanged atmospheric absorption we made back then (see your general comment and comment on L79-81) the problem was linear behaved. If a represents an old state, b a new state and c a factor that applies to both, $c(b-a) = cb + ca$, i.e. it does not matter whether the factor is applied to the original value or the perturbation. However, neither this assumption nor the explanation is needed any more in the revised manuscript, which uses kernels which have been explicitly determined by previous authors for perturbations (and applied to the delta of albedo in our revised methodology).

Figure S5: This figure appears twice in a row, and “radiation” in the x-axis label is misspelled in both. Also, the captions differ.

Sorry! Now obsolete because the whole section 1.3 was changed due to the methodology revision. There were no different captions, but the caption of one appearance was wrongly formatted and thus indistinguishable from the following main text.

Section 1.4: I get completely lost in this section. First off, what is point that is trying to be made here about the “relation between shortwave and net total radiation”? Perhaps it would be helpful to ground this text in the radiative forcing literature? The authors seem to be alluding to differences between instantaneous shortwave radiative forcings and effective radiative forcings and the importance of atmospheric adjustments, but this is not made at all clear, and leaves me guessing because – like my previous comment – the section seems to be describing physics in terms of state variables.

Total net radiation is the sum / net effect of shortwave and longwave net radiation. We expected a partly compensating effect of longwave radiation because e.g. a surface with a lower albedo typically develops a higher temperature and thus larger long-wave emission partly compensating the increased net downward shortwave flux. The section is not needed any more with the revised methodology, and the revised and hopefully clearer section 1.3 now includes an effort to align direct and indirect effects in our framework with the concept of IRF vs. ERF.

L147-153: I cannot follow a single methodological description here. What is “offset” and “reduced major axis”? Are the authors regressing SW vs. NETRAD, and somehow utilizing the slope and intercept terms? Why not show this figure and the robustness of this relationship? What are the “site-wise resulting factors” that are being shown in Fig. S6 – are these the slopes? And are they statistically meaningful?

At this point of the original submission of the supplementary material, the synonymity between “offset” and “intercept”, as well as of “reduced major axis” and “type II regression” had already been established upon first occurrence (L71-74 of the original supplement). However, both occurrences have become obsolete by our revised, kernel-based methodology. Where the same technique is used now, we follow your suggestion to prefer “intercept” over “offset”, always show the corresponding scatterplot, and explain the reduced major axis in more detail including a literature reference (e.g. section 1.3/ Fig. S4: “All regression-like empirical linear equations are based on the reduced major axis¹³ technique, a subtype of what is sometimes referred to as model II regression in which the x and y

variable have unknown errors and are interchangeable such that equation $x(y)$ is the inverse of $y(x)$.” (Supplement L122). Note that while we are convinced that it is more precise to apply reduced major axis instead of ordinary regression in this case, the difference would have been very small due to the high R^2 values, and these regressions are not used to produce any results given in the main article, but only for diagnosing differences between the kernels in the supplement.

Section 1.5: Either include the equation(s) for this calculation or refer the reader to the specific equations published elsewhere.

Done, the table number in Myhre (1998) and Equation numbers in Ney et al. (2019) are now both referred to: “We converted cumulative CO_2 uptake or loss to radiative forcing following the simple scheme by Myhre, et al. ¹⁵ (Table 3 therein) assuming an instant effect of the airborne fraction, and a linearized radiative forcing (RF) response to concentration change as also described e.g. in Ney, et al. ¹⁶ (Eqn. 5-6 therein), but using updated values for the airborne fraction (0.44 ¹⁷) and CO_2 concentration (420 ppm)” (Supplement L143). The same was done in the main article.

L170: What are “difference forcings”? And do the authors mean “business-as-usual” – not “business and usual”?

Apologies for the typo, “business-as-usual” is correct. The unusual term “difference forcings” is now avoided, and the sentence further changed due to advice by Reviewer 3 to avoid the term “radiative forcing” in a context of albedo changes (see Reviewer 3 comment 4): “Radiative effects of both, CO_2 uptake and albedo, are expressed as differences in top-of-atmosphere net radiation compared to a “business-as-usual” scenario ...” (Supplement L150).

L175-177: This sentence is hard to follow. What are “climate-change motivated forms of land use change”?

Sentence changed to: “The analysis does not aim at detailed predictions, but at revealing the main differences between land use and management changes aiming to increase NEP, albedo, or both” (L155).

Fig. S7: “delta RF” in y-labels – where is this defined?

We changed RF to R according to comment 4 of Reviewer 3 and define in the caption (new figure number S5): “ ΔR : global top-of-atmosphere net radiation change resulting from combined CO_2 radiative forcing and surface albedo effect per each % of land surface on which land use is changed according to the respective scenario.”

Reviewer #2 (Remarks to the Author):

In this manuscript, the authors discussed the co-variability of net ecosystem productivity (NEP) and surface albedo due to land use change in the context of climate change mitigation options. First, they analyzed observations from 176 flux sites and found that NEP and albedo cannot be changed independently. Then, they estimated the time series of radiative forcing for different hypothetical land-use scenarios and found a scenario with a cooling effect in both the short- and long-term.

The manuscript is well-written and contains valuable information for considering climate change mitigation options. Therefore, I recommend the publication after minor modifications and clarifications.

Thanks!

Specific comments:

Fig. 2: Can you explain the unit of the y-axis of the lower panels? What does “per % land surface” mean?

The respective part of the caption was changed to: “e)-f): Resulting pathways of global top-of-atmosphere net radiation change, ΔR , per each % of land surface on which land use is changed according to the panel above. The ensemble mean (bold red line) and full uncertainty range (shade) result...” (L188). The y axis title was shortened to focus on this caption description and RF was replaced by R due to comment 4 of Reviewer 3.

L426: If there are multi-year observations, how did you estimate the initial state of the albedo?

Multi-year observations were first aggregated to the NEP, albedo and (in the revised methodology) monthly albedos for each site by averaging across all available years, before entering the scenario modelling exercise. We now specified in the methods part of the main article: “Scenario 1 (NEP optimization): The site with the highest (multi-) annual average NEP. Scenario 2 (albedo optimization regardless of NEP): The site with the highest (multi-) annual average α_s .” (L467), and in the supplementary material (caption of new Fig. S4): “Each data point corresponds to a climatological site-month, i.e. in case of multiple years of available measurements or modelling results, they were averaged across years but separately for each site and month.” (Supplement L118).

L471: How did you evaluate the representative area of the observation sites and connect them to the global radiative forcing?

We considered change occurring to only a small percentage of the total land surface (e.g. 1 % for the unit of ΔR , see reply to comment on Fig. 2, scaling up to 10% already would create radiative effects similar to other RFs). It was therefore assumed that for each site, enough changeable land surface can be found representing its respective land use, soil and climate conditions. We clarify in the strongly revised methods subsection ‘Mitigation scenarios’: “The total global effect was computed assuming that the mean of the radiative change caused at all sites happens on 1 % of the global land surface.” (L490). An alternative strategy where sites are weighed according the fraction of land surface they represent would be possible, but given the multiple criteria (land cover type and climate at least) would introduce an additional assumption-based methodology to document in the publication, which did not appear reasonable to us given the conceptual nature of the question (comparing the roles of albedo and NEP in net radiative change rather than making exact predictions).

L477: I am not fully convinced by the method that calculates the top of the atmosphere net shortwave radiation by multiplying a constant by the surface net shortwave radiation. It may be necessary to explain why this method was chosen and how much uncertainty it has compared to other methods (e.g., radiative kernels as described in Pendergrass et al. (2018) cited by the authors).

We now indeed use radiative kernels, also since similar comments by Reviewer 1 (general comments) and Reviewer 3 (comment 3) demonstrate that our original approach was not convincing. Interestingly, however, the extensive comparison of four kernel datasets, their ensemble mean, and measured SWin at the surface to each other (new supplement section 1.3 / Fig. S4) and the resulting small changes in the main manuscript (Fig. 2) demonstrate that the old methodology was very successful in predicting the ensemble mean and variance which are now determined directly based on kernels.

Fig. S1: It may be helpful to the reader if the IGBP type for each site is shown in the figure.

Done.

Fig. S2: Please add a unit to the x-axis.

Done (°C).

Fig. S5: There are two same figures in the word file.

Apologies. The figure does not exist any more due to the methodology revisions following your comment L477, comment 3 of Reviewer 3, and the ERF comment of Reviewer 1.

Supplementary material L88: The reference is missing in my environment.

Was a reference to the next figure, obsolete now due to changing the whole section (see above).

Supplementary material L213: Please add an explanation for what NEP_{pot} means.

We added a comma, Eqn number and the explanation: “where NEP_{pot} is the net ecosystem production before considering any saturation effects.” (Supplement L194).

Reviewer #3 (Remarks to the Author):

Review, Graf et al: Co-variability of Land Surface Albedo and Carbon Uptake: Opportunities and Limitations for Climate Change Mitigation

In this manuscript, the authors study the relationships between surface albedo and CO₂ uptake across a wide set of in situ observations from a variety of station networks, primarily FLUXNET / ICOS, and AMERIFLUX. The aim is to ascertain whether or not climate change mitigation through active modulation of vegetated land cover to increase either CO₂ uptake or surface albedo (to cause climatic cooling) is likely to cause enhancing or hindering changes in the other variable. The primary answer seems to be that an expected & significant negative relationship exists between the two, and that thus far unavailable technological means would be needed to create vegetation species with both high uptake capacity and high surface albedo if cooling potential were to be maximized through this process.

The manuscript is topically appropriate for Communications Earth & Environment and the text generally flows quite well, delivering its message in a clear way. Its wider practical impact would lie in setting boundary conditions for possible future geoengineering efforts through vegetation modulation, which in itself is a useful addition to the present literature.

Thank you.

However, certain aspects of the calculations behind the results are not yet justified to this reviewer's satisfaction and require revision and possible reconsideration before publication may be considered. These comments are listed below, with major comments listed first.

See reply to each comment.

Major comments:

1. lines 202-205: In general the statement that PAR majority is not used for photosynthesis is likely so. However, the conversion used here, based on measurements by Black et al. (1996), was based on a single Aspen forest site for a limited time. What is the robustness of the conversion and associated analysis as shown here? Also, e.g. Chen et al. (2008) state that cloudiness variability may impact photosynthetic energy-use efficiency markedly. Given 100-yr simulations presented here, has there been considerations of potential shifts in e.g. cloudiness as a source of uncertainty?

Apologies, re-reading we realize that (possibly because of not taking care during an earlier shortening) the sentence before this suggests the interpretation that the cited conversion factor is a site-specific measurement result directly relating photosynthetic energy use to PAR, which it is not. It is rather a chemical constant, used with little variation in many papers, relating moles of sequestered CO₂ to the energy required for it. Only the 2.7% mentioned later indeed relates energy use to PAR, but none of both factors are needed or used in any way for the 100-yr simulations. We now clarified the introduction of the two factors and included information on the robustness (maximum, mean, median and standard deviation) as follows: "In our dataset, this fraction could be estimated by applying the energy intensity of photosynthesis ($0.469 \text{ J } \mu\text{mol}^{-1} \text{ CO}_2^{34,35}$) to gross primary productivity GPP and comparing to non-reflected incoming shortwave radiation SW_{net} . While the resulting fraction of radiation energy used for photosynthesis can vary depending on ecosystem and cloudiness³⁶, its site-averaged long-term value was nowhere above 2.7 % and on average (both arithmetic mean and median) 1.2 % (standard deviation 0.6 %)." (L222). We included Black et al. (1996) as primary source

of the fluxes used in Blanken et al. (1997) (references 34 and 35) as well as Chen et al. (1999) as a mentioning of the variability of photosynthetic energy use with cloudiness (reference 36).

2. Methods – mitigation scenarios: Given that surface albedo depends not only on the optical properties of the surface but also on the directionality of the incoming shortwave radiation, the discussion around surface albedo seems to also omit considerations of cloudiness variability which can and will alter the balance between direct and diffuse irradiance and thus change the blue-sky albedo considered here. Given the changes made in the mitigation scenarios, particularly in #2, did the authors also consider similarity in illumination as one of the partner site selection conditions?

Note that the current albedo is not a blue-sky albedo, but each site's genuine (effective, resulting from the site's varying solar angle and cloudiness properties) albedo. Even so it is still a good point, but given the status quo of available measurements at each site, making the direct to diffuse ratio mandatory would either reduce the site dataset to a fraction or require lengthy introduction of a proxy (e.g. using the ratio measured to potential SW_{in} as a measure of cloudiness). Given the fact that a comparison of our old (corrected SW_{in} based) method to the new (kernel-based) one revealed a remarkably good agreement (see next comment) and thus does not support the expectation of strong effects of cloudiness changes, we would prefer keeping the methodology straightforward over adding another constraint with small effect. Note that the new methodology uses a monthly (as opposed to annual in the original submission) time step to compute the effect of radiation and site-specific albedo kernels, which better reflects the state of the art in modelling albedo-related effects and implicitly includes at least some of the spatiotemporal variability and cloudiness, solar angle and resulting direct/diffuse ratio.

3. Methods – net radiative forcing: It was not yet fully clear how the radiative forcing was calculated for the mitigation scenarios, or if the calculation is now precise enough. It seems that a static constant is used for all sites to convert radiative energy budget changes at surface to those at TOA. Given that there are published radiative kernels (from satellite data with global coverage) which yield the TOA radiative perturbation per unit change in surface albedo (for example, Bright & O'Halloran, 2019), why were these not used instead?

We now indeed use radiative kernels (including Bright & O'Halloran 2019), also given that similar comments by Reviewer 1 (on instant vs. effective radiative forcing) and Reviewer 2 (comment L477) demonstrate that our original approach was not convincing. Interestingly, however, the extensive comparison of four kernel datasets, their ensemble mean, and measured SW_{in} at the surface to each other (new supplement section 1.3 / Fig. S4) and the resulting small changes in the main manuscript (Fig. 2) demonstrate that the old methodology was very successful in predicting the ensemble mean and variance which are now determined directly based on kernels.

4. General: Also, as a cautionary note on terminology, 'radiative forcing' is currently not a favored term when discussing radiative energy balance changes due to surface albedo changes, as the driver is not seen as external to the system (see e.g. IPCC glossaries). Surface albedo feedback (SAF) is currently preferred. Please reconsider this part of the terminology.

We now avoid using the term 'radiative forcing' where discussing radiative energy balance changes due to surface albedo changes (while keeping it for CO_2 -based changes), except for a few new (mostly following comments of Reviewer 1) references in the supplementary material, where RF was explicitly used in the context of surface albedo changes. Not using the term at all would make it hard for readers to connect our terms to the original literature.

However, as another matter of precaution, we did neither adopt the term 'surface albedo feedback'. The reason is that a search led us to believe that the term is mostly (if not almost exclusively) used in cases where surface albedo changes in response to climate change (true feedback). In our study, in contrast, surface albedo change due to anthropogenic land-use change is considered, which is not a 'feedback' in the sense of the abovementioned usage of 'surface albedo feedback'. We therefore resort to terms like 'albedo effect'.

Minor comments by line numbers:

106-107: "to violate a safe distance to tipping points" is a rather unclear expression, revise for clarity.

Changed to: "Global emissions from fossil-fuel burning are still increasing and by the mid-21st century greenhouse-gas concentrations are likely higher than recommended to safely avoid tipping points ¹³" (L104).

154-155: The negative relationship is intuitively quite clear given that the surfaces with highest natural albedos, snow/ice and desert sand, are both rather incapable of sustaining carbon-capturing vegetation. Suggest that the authors already refer to the upcoming deeper analysis on this in the section on "Evidence".

*Good point (see also first editor comment), we changed and extended this end of the first section as follows (and deleted the first sentence of the following section): "In general, the fact that a high albedo and large NEP are often incompatible is documented ²⁰ and may be expected from the fact that plant covers have a lower albedo than most natural unvegetated surfaces. However, to our knowledge the global nature, beyond forest effects, of this hypothetically natural relationship has not been demonstrated, and its shape as seen in **Error! Reference source not found.** not quantified. Our findings further suggest it applies as an envelope to maximum NEP and α_s values, but not to the bulk of the examined, and mostly economically used, sites. This suggests much of the land surface can, and possibly did, provide better climate services than under current management by having an NEP and α_s combination closer to their joint natural limit. We conceptually examine this possibility in the following section. Because of potential seasonal co-variability of α_s and irradiation this will be done with monthly albedo observations ³⁰, the role of which in causing our findings will further be examined in a further subsection." (L158).*

260 and elsewhere: Some inconsistent citation styles are used in the manuscript, please harmonize.

We corrected the style in the Lugato citation and also scanned, and corrected where necessary, the remaining text. However, due to an editor suggestion to match the formal requirements for a final version already in this revision, citations are now number-based.

Conclusions: It is worth while to note somewhere in the text that the focus here is on (vegetated) land surfaces. Albedo modification of oceans via e.g. enhanced or artificial sea ice generation would globally fall into the second category, but its impact is not considered in the manuscript.

Good point, we added „of the land surface" to the first sentence of the conclusion (L351).

Citations:

Black, T. A., Den Hartog, G., Neumann, H. H., Blanken, P. D., Yang, P. C., Russell, C., ... & Novak, M. D.

(1996). Annual cycles of water vapour and carbon dioxide fluxes in and above a boreal aspen forest. *Global Change Biology*, 2(3), 219-229.

Bright, R. M., & O'Halloran, T. L. (2019). Developing a monthly radiative kernel for surface albedo change from satellite climatologies of Earth's shortwave radiation budget: CACK v1. 0. *Geoscientific Model Development*, 12(9), 3975-3990.

Chen, W. J., Black, T. A., Yang, P. C., Barr, A. G., Neumann, H. H., Nesic, Z., ... & Cuenca, R. (1999). Effects of climatic variability on the annual carbon sequestration by a boreal aspen forest. *Global Change Biology*, 5(1), 41-53.

Decision letter and referee reports: second round

2nd Jun 23

Dear Dr Graf,

Your manuscript titled "Co-variability of Land Surface Albedo and Net Carbon Uptake leaves room for Climate Mitigation" has now been seen by 2 reviewers, and I include their comments at the end of this message. They find your work of interest, but some important points are raised.

It seems to us that a deadlock has been reached between your arguments and reviewer #1's, and we will therefore seek independent advice from an adjudicating referee.

But before we do so, please revise your manuscript in response to the remaining concerns and supply us with a point-by-point response to referee #1's report. We will send your responses, the report and the revised manuscript to the adjudicator.

We would also appreciate suggestions for adjudicators that you have not collaborated with over the past 5 years or so, and who you would trust to be fair as well as competent to evaluate your disagreement with the reviewer. Please supply at least six names — we cannot promise to use any of them, but we will do so if it seems possible and appropriate.

Please use the following link to submit your revised manuscript, point-by-point response to the referees' comments (which should be in a separate document to any cover letter) and the completed checklist:

[Link Redacted]

We hope to receive your revised paper within six weeks; please let us know if you aren't able to submit it within this time so that we can discuss how best to proceed. If we don't hear from you, and the revision process takes significantly longer, we may close your file. In this event, we will still be happy to reconsider your paper at a later date, as long as nothing similar has been accepted for publication at Communications Earth & Environment or published elsewhere in the meantime.

We understand that due to the current global situation, the time required for revision may be longer than usual. We would appreciate it if you could keep us informed about an estimated timescale for resubmission, to facilitate our planning. Of course, if you are unable to estimate, we are happy to accommodate necessary extensions nevertheless.

Please do not hesitate to contact me if you have any questions or would like to discuss these revisions further. We look forward to seeing the revised manuscript and thank you for the opportunity to review your work.

Best regards,

Jinfeng Chang, PhD
Editorial Board Member
Communications Earth & Environment
orcid.org/0000-0003-4463-7778

Alienor Lavergne, PhD
Associate Editor
Communications Earth & Environment

EDITORIAL POLICIES AND FORMATTING

Editorial Policy: [Policy requirements](https://www.nature.com/documents/nr-editorial-policy-checklist.pdf) (Download the link to your computer as a PDF.)

Furthermore, please align your manuscript with our format requirements, which are summarized on the following checklist:

[Communications Earth & Environment formatting checklist](https://www.nature.com/documents/commsj-phys-style-formatting-checklist-article.pdf)

and also in our style and formatting guide [Communications Earth & Environment formatting guide](https://www.nature.com/documents/commsj-phys-style-formatting-guide-accept.pdf)

.

***** DATA:** Communications Earth & Environment endorses the principles of the Enabling FAIR data project (<http://www.copdess.org/enabling-fair-data-project/>). We ask authors to make the data that support their conclusions available in permanent, publically accessible data repositories. (Please contact the editor if you are unable to make your data available).

All Communications Earth & Environment manuscripts must include a section titled "Data Availability" at the end of the Methods section or main text (if no Methods). More information on this policy, is available at <http://www.nature.com/authors/policies/data/data-availability-statements-data-citations.pdf>.

If a community resource is unavailable, data can be submitted to generalist repositories such as [figshare](https://figshare.com/) or [Dryad Digital Repository](http://datadryad.org/). Please provide a unique identifier for the data (for example a DOI or a permanent URL) in the data availability statement, if possible. If the repository does not provide identifiers, we encourage authors to supply the search terms that will return the data. For data that have been

obtained from publically available sources, please provide a URL and the specific data product name in the data availability statement. Data with a DOI should be further cited in the methods reference section.

Please refer to our data policies at http://www.nature.com/authors/policies/availability.html.

REVIEWER COMMENTS:

Reviewer #1 (Remarks to the Author):

Please see .pdf attachement.

Reviewer #3 (Remarks to the Author):

The authors have made a substantial effort to revise the manuscript to address my concerns. Particularly the use of multi-source radiative kernels to estimate the TOA radiative flux perturbations is in my view a much more justifiable method.

Further revisions to the text have in my view also succeeded in better justifying the achieved results and acknowledging the shortfalls/uncertainties where they occur. As a result, and acknowledging the sufficient novelty value of the study, I can now recommend publication in Communications Earth & Environment.

Review of COMMSENV-22-0963A

The revision has elevated the quality of the manuscript with regards to improved readability (improved writing) and decreased sloppiness levels (reference and text formatting, labeling, figure redundancy issues etc.).

My biggest criticism of the original manuscript surrounded the lack of attention given to the atmospheric adjustments following forest cover change. This remains my biggest criticism. ERF (as IRF + adjustments) is the best indicator of climate change (see IPCC AR6 Chapter 7), and the models clearly point to the relevance of the atmospheric adjustments. Look (again) at Table 7 in Smith *et al.* (2020) where the mean adjustment across the 12 included CMIP6 models offset IRF by ~43% (i.e., $-0.08 = -0.14 + 0.06 \text{ W/m}^2$). Yes, these numbers are based on land use/land cover change forcings (not just forest cover change) and are highly uncertain, but one cannot deny their importance as is suggested in your reply:

“...we do not agree that Table 7 in Smith et al. 2020 demonstrates a large discrepancy between albedo change based IRFs and ERFs”

While the effort to employ radiative kernels for surface albedo change in the revision is commendable, the authors seem to have misunderstood what these kernels (i.e., refs 67, 82-84) represent as they do **NOT** account for atmospheric adjustments (“corrections”) as is claimed on L111-113 of the Supporting Information:

“The kernel has the same unit, approximate magnitude and spatiotemporal variation as SW_{in} , but implicitly includes corrections for all or some effects of ΔSW_{in} and ϵ , assuming that both depend linearly on $\Delta\alpha$.”

And on L523 – 529 of the main article:

“A part of these effects can be included in model-based surface albedo kernels⁸²...”

And on L222 of the Supporting Information:

“Top-of-atmosphere net radiative effect of albedo changes”

Such statements are erroneous, as the applied albedo change kernels by their very design account **only** for the instantaneous (shortwave) radiative perturbation at TOA (i.e., the IRF).

Adjustments following CO₂ perturbations (NEP changes) are also ignored in the study where the radiative forcing calculation is based on SARF (i.e., Myhre *et al.* 1998). Including these would have the effect of increasing CO₂'s RF by ~5%. See Table 7.4 of Chapter 7 in IPCC AR6 WG1.

While I understand the (biogeophysically-mediated) atmospheric adjustments cannot be quantified in the study, the authors owe it to the readers to be honest about the limitations and caveats of their methods and at least discuss qualitatively the relevance of the adjustments in different forest biomes or geographic regions – and how they might influence their scenario results. There is ample empirical evidence to support such a discussion, and I provide references below. For example, forests in many boreal and temperate regions are seen to enhance convective-driven low-level cloud cover (Xu *et al.* 2022), and such optically-thick clouds are seen to have a net TOA radiative cooling effect (L'Ecuyer *et al.*, 2019). Thus, positive IRFs from decreased surface albedo changes linked to an enhanced forest cover in these regions are likely to be offset by cloud-related atmosphere adjustments, likely resulting in biogeophysical ERFs that are lower in magnitude than the surface albedo change-only IRF. Taken together with adjustments following CO₂ concentration changes (or enhanced NEP), the net combined TOA forcing is likely lower than that shown in the bottom panels of Figure 2.

I find it very difficult to support the publication of this manuscript in an esteemed journal like *Nature Communications Earth & Environment* until the erroneous methodological statements are remedied and the authors acknowledge the study's caveats and limitations, including an honest discussion about how these affect their main results and conclusions.

REFERENCES:

L'Ecuyer, T. S., Y. Hang, A. V. Matus, and Z. Wang (2019), Reassessing the Effect of Cloud Type on Earth's Energy Balance in the Age of Active Spaceborne Observations. Part I: Top of Atmosphere and Surface, *Journal of Climate*, 32(19), 6197-6217.

Xu, R., et al. (2022), Contrasting impacts of forests on cloud cover based on satellite observations, *Nature communications*, 13(1), 670.

Dear Editors and Reviewer,

please find below our point-to-point responses to all Reviewer comments.

REVIEWER COMMENTS:

Reviewer #1 (Remarks to the Author):

The revision has elevated the quality of the manuscript with regards to improved readability (improved writing) and decreased sloppiness levels (reference and text formatting, labeling, figure redundancy issues etc.).

Thank you very much.

My biggest criticism of the original manuscript surrounded the lack of attention given to the atmospheric adjustments following forest cover change. This remains my biggest criticism. ERF (as IRF + adjustments) is the best indicator of climate change (see IPCC AR6 Chapter 7), and the models clearly point to the relevance of the atmospheric adjustments.

We agree with the reviewer, apologize for our misunderstanding on the capability of kernels to solve the issue, and discuss the direction and magnitude of adjustments in the second revision as suggested.

Look (again) at Table 7 in Smith et al. (2020) where the mean adjustment across the 12 included CMIP6 models offset IRF by ~43% (i.e., $-0.08 = -0.14 + 0.06 \text{ W/m}^2$). Yes, these numbers are based on land use/land cover change forcings (not just forest cover change) and are highly uncertain, but one cannot deny their importance as is suggested in your reply:

“...we do not agree that Table 7 in Smith et al. 2020 demonstrates a large discrepancy between albedo change based IRFs and ERFs”

We agree that our wording in this reply (which was not part of the revised MS) was poorly chosen given the different expectations the word “large” may raise. The reply was on your earlier comment “Model intercomparison studies have shown that the atmospheric adjustments can either completely offset the albedo change-driven iRF or reinforce it further (c.f., Table 7 in Smith et al. 2020).” Only 2 out of 14 models exhibit such a complete offset, and they were the only ones in the table with no negative IRF, GISS-E2-1-G p1 and NorESM2-LM. In all other cases $|Adj|$ was smaller than $|IRF|$. The mean statistics across the table according to our calculations, once including all 14 and once excluding NorESM2-LM (as done by Smith et al. at the top of the right column of p 9608), were:

	Percentage difference (mean Adj / mean IRF)	R^2	Regression slope IRF vs. ERF (as reference / x variable)	Slope, forced through origin
All 14 models	36 %	0.69	0.73	0.77
w/o NorESM2-LM	22 %	0.83	1.11	1.20

Given the large differences (including in sign) between models and our initial worst-case expectation from your earlier comment that adjustments might generally make IRF meaningless, we concluded after this analysis that the table does not (convincingly) demonstrate this type of large discrepancy. However, we fully agree that an inter-model mean deviation of ERF from IRF of 22%, 36% or even (as your comment above suggests) 43% should definitely be discussed as a source of systematic

uncertainty in our manuscript. Note that we would need further guidance from your side to mention the figure of 43%, since we are currently unable to reproduce it. Above you write “12 included CMIP6 models”, but the table contains 14 models, all of which, according to the methods section of Smith et al. (2020) were involved in CMIP6 (p 9593). The authors do seem to suggest omitting NorESM2-LM from the inter-model mean (top of the right column of p 9608), which would decrease the percentage difference even further to 22 % (see table above), but we could not find any other guidance in the paper to omit models and arrive at the exact figures in your above comment. Including all models as done at the bottom of the table, your abovementioned calculation “(i.e., $-0.08 = -0.14 + 0.06 W/m^2$)” would to our current best knowledge thus have to be revised “(i.e., $-0.09 = -0.14 + 0.05 W/m^2$), which results in a 36% relative difference. Possibly, you have access to original raw data while the published table may contain round-off or other errors, or can provide guidance which 12 out of the 14 models to pick. Without such guidance, we suggest to use the figure 36% (lower than your suggested percentage but higher than the one without NorESM2-LM) when discussing ERF-related uncertainty in our revised manuscript. The new text parts where the percentage is mentioned are cited below.

While the effort to employ radiative kernels for surface albedo change in the revision is commendable, the authors seem to have misunderstood what these kernels (i.e., refs 67, 82-84) represent as they do NOT account for atmospheric adjustments (“corrections”) as is claimed on L111-113 of the Supporting Information:

“The kernel has the same unit, approximate magnitude and spatiotemporal variation as SW_{in} , but implicitly includes corrections for all or some effects of ΔSW_{in} and ϵ , assuming that both depend linearly on $\Delta\alpha$.”

Changed to (L110): “The kernel has the same unit, approximate magnitude and spatiotemporal variation as SW_{in} . Unlike measured SW_{in} at the surface it does not require a correction for static atmospheric absorption, and unlike SW_{in} at the TOA it does not require a correction for static surface albedo masking by clouds. However, the methodology to derive kernels does not include the atmospheric adjustments in which ERF differs from IRF. With presently available model outcomes, we can only present a rough estimate of these additional effects. Assuming that a comparison of ERF and IRF of land-use change across 14 CMIP6 models¹¹ is dominated by albedo and CO_2 sequestration changes, the inter-model mean relative difference of - 36 % (Table 7 in¹¹, ERF less important than IRF) can serve as a rough indicator of a possible overestimation in our results. It should be noted, however, that the individual models contributing to this study saw adjustments of both positive and negative signs, with a standard deviation between models ($0.08 W m^{-2}$) larger than the inter-model mean adjustment ($0.05 W m^{-2}$)”.¹¹ refers to Smith et al. 2020.

And on L523 – 529 of the main article:

“A part of these effects can be included in model-based surface albedo kernels 82...”

Now replaced by separate paragraphs on the issues the kernels can solve, and those they cannot. At the mentioned point, we continue (L535): “In our case, SW_{in} and $\Delta\alpha$ are measured at the land surface, while effects at the TOA are of interest for comparison to CO_2 effects. This would require an additional correction either for cloud masking of the surface (if using SW_{in} at TOA), or for atmospheric absorption (if using $SW_{in}\Delta\alpha_{in}$ at the surface). We therefore replace SW_{in} by model-derived, publicly available surface albedo kernels⁸⁵”. We then continue with the last version of the kernel description and conclude with a new paragraph (L572): “The final net effect of surface albedo further differs from kernel-based instantaneous shortwave effects at the TOA, because any land use change modifies surface temperature, dust emission⁷⁰, sensible and latent heat fluxes and resulting atmospheric temperature, humidity, particle and cloud profiles, which in turn affect SW_{in} and longwave radiation.

Quantification of the resulting effective radiative forcing (ERF⁶⁹) requires a global circulation model run with the exact spatiotemporal pattern and magnitude of all surface property changes accompanying the albedo change for each scenario, and is thus missing in most sensitivity studies on land surface albedo changes. To provide a rough estimate of the importance of the adjustments distinguishing ERF from IRF, we refer to a study comparing among others the net effect of the same land use change across 14 CMIP6 models³¹.³¹ refers to Smith et al. 2020,⁶⁹ to Sherwood et al. 2015,⁷⁰ to Andrews et al. 2017.

And on L222 of the Supporting Information:

“Top-of-atmosphere net radiative effect of albedo changes”

Such statements are erroneous, as the applied albedo change kernels by their very design account only for the instantaneous (shortwave) radiative perturbation at TOA (i.e., the IRF).

Changed to (L231): “Instantaneous top-of-atmosphere shortwave radiative effect of albedo changes”. We also appended to the end of the same paragraph: “The net effect of albedo changes after atmospheric adjustments and across all wavelengths is subject to further, large uncertainties, which we discuss (section 1.3 and main article) based on the current state of knowledge”.

Adjustments following CO₂ perturbations (NEP changes) are also ignored in the study where the radiative forcing calculation is based on SARF (i.e., Myhre et al. 1998). Including these would have the effect of increasing CO₂'s RF by ~5%. See Table 7.4 of Chapter 7 in IPCC AR6 WG1.

We appended to the main article methods section “NEP-related radiative forcings in scenarios” (L527): “Tropospheric adjustments can further reinforce the effective radiative forcing of CO₂ by approximately 5 % (¹⁵, Table 7.4 therein). This effect is not included in our computations, since current availability of model-based evidence would not allow us to consistently apply the same type of correction to albedo effects (following subsection).”¹⁵ refers to IPCC AR6 WG1.

While I understand the (biogeophysically-mediated) atmospheric adjustments cannot be quantified in the study, the authors owe it to the readers to be honest about the limitations and caveats of their methods and at least discuss qualitatively the relevance of the adjustments in different forest biomes or geographic regions – and how they might influence their scenario results. There is ample empirical evidence to support such a discussion, and I provide references below. For example, forests in many boreal and temperate regions are seen to enhance convective-driven low-level cloud cover (Xu et al. 2022), and such optically-thick clouds are seen to have a net TOA radiative cooling effect (L'Ecuyer et al., 2019). Thus, positive IRFs from decreased surface albedo changes linked to an enhanced forest cover in these regions are likely to be offset by cloud-related atmosphere adjustments, likely resulting in biogeophysical ERFs that are lower in magnitude than the surface albedo change-only IRF. Taken together with adjustments following CO₂ concentration changes (or enhanced NEP), the net combined TOA forcing is likely lower than that shown in the bottom panels of Figure 2.

We thank the reviewer for these constructive suggestions, according to which we have added the following discussion paragraph to the results and discussion section of the main article, subsection “Different scenarios for mitigating climate change” (after Fig. 2 at L212): “Further uncertainties, which we cannot quantify with currently available model-based evidence for our specific scenarios, include the atmospheric adjustments (¹⁵, therein chapter 7.3) and feedbacks (chapter 7.4) following land-use change. An intercomparison of effective (ERF) vs. instantaneous radiative forcing (IRF) of land-use change across CMIP6 models³¹ exhibited both reinforcing and offsetting adjustment effects depending on model. Their inter-model mean importance of -36 % of IRF suggests a corresponding overestimation of net radiation change in Fig. 2. A reduction of this magnitude is plausible given that

*e.g. forests mostly stimulated low level clouds in boreal and temperate regions, which dominate our dataset, in a recent study³². The dominant stimulated cloud types have net cooling effects at the TOA³³ and would thus counteract the heating effect of lower forest albedo. However, the forest breeze circulation contributing to such cloud stimulation³⁴ can also reduce cloudiness over nearby non-forest surfaces. Assuming that albedo strongly contributes to the land-use effects inhibited according to Smith, et al.³¹, while the effect of CO₂ alone has been shown to be slightly reinforced by tropospheric adjustments by approximately 5 %⁽¹⁵⁾, therein table 7.4), adjustments can also affect the balance between albedo and CO₂ effects seen in Fig. 2. A weakened albedo and reinforced CO₂ effect would underline our finding that pure albedo optimization (SC2) is prone to result in long-term warming, and suggest an even earlier onset of net cooling for NEP optimization (SC1).”¹⁵ refers to IPCC AR6 WG1,³¹ to Smith et al. 2020,³² to Xu et al. 2022,³³ to L’Ecuyer et al. 2019 and³⁴ to Teuling, A. J. et al. Observational evidence for cloud cover enhancement over western European forests. *Nature Communications* 8 (2017). <https://doi.org:10.1038/ncomms14065>.*

I find it very difficult to support the publication of this manuscript in an esteemed journal like Nature Communications Earth & Environment until the erroneous methodological statements are remedied and the authors acknowledge the study’s caveats and limitations, including an honest discussion about how these affect their main results and conclusions.

We believe that we have now clarified the issues pointed out by the reviewer and better explained the caveats and limitation of the results, and we thank the reviewer for pointing them out.

REFERENCES:

L’Ecuyer, T. S., Y. Hang, A. V. Matus, and Z. Wang (2019), Reassessing the Effect of Cloud Type on Earth’s Energy Balance in the Age of Active Spaceborne Observations. Part I: Top of Atmosphere and Surface, *Journal of Climate*, 32(19), 6197-6217.

Xu, R., et al. (2022), Contrasting impacts of forests on cloud cover based on satellite observations, *Nature communications*, 13(1), 670.

In addition to the above changes we added near the end of the conclusions section, directly after the existing discussion of the importance of ERF (L419): “A future larger body of land-use change scenarios for which ERF is modelled, and converging results between improved models would enhance the estimation of adjustment effects in studies not involving dedicated general circulation model runs.”

Decision letter and referee reports: third round

11th Jul 23

Dear Dr Graf,

Your manuscript titled "Co-variability of Land Surface Albedo and Net Carbon Uptake leaves room for Climate Change Mitigation" has now been seen by our reviewers, whose comments appear below. In light of their advice we are delighted to say that we are happy, in principle, to publish a suitably revised version in Communications Earth & Environment under the open access CC BY license (Creative Commons Attribution v4.0 International License).

We therefore invite you to revise your paper one last time to address the remaining concerns of our reviewers. At the same time we ask that you edit your manuscript to comply with our format requirements and to maximise the accessibility and therefore the impact of your work.

EDITORIAL REQUESTS:

*****Please take care to match our formatting and policy requirements. We will check revised manuscript and return manuscripts that do not comply. Such requests will lead to delays. *****

SUBMISSION INFORMATION:

OPEN ACCESS:

Communications Earth & Environment is a fully open access journal. Articles are made freely accessible on publication under a [CC BY license](http://creativecommons.org/licenses/by/4.0) (Creative Commons Attribution 4.0 International License). This license allows maximum dissemination and re-use of open access materials and is preferred by many research funding bodies.

For further information about article processing charges, open access funding, and advice and support from Nature Research, please visit <https://www.nature.com/commsenv/article-processing-charges>

At acceptance, you will be provided with instructions for completing this CC BY license on behalf of all authors. This grants us the necessary permissions to publish your paper. Additionally, you will be asked to declare that all required third party permissions have been obtained, and to provide billing information in order to pay the article-processing charge (APC).

[Link Redacted]

Best regards,

Jinfeng Chang, PhD
Editorial Board Member
Communications Earth & Environment
orcid.org/0000-0003-4463-7778

Alienor Lavergne, PhD
Associate Editor
Communications Earth & Environment

REVIEWERS' COMMENTS:

Reviewer #1 (Remarks to the Author):

Please see attachment.

Reviewer #1 comments:

Review of COMMSENV-22-0963B

I thank the authors for their efforts in addressing my biggest concerns. They demonstrated that they invested effort in becoming acquainted with the recommended literature and successfully extracted and integrated the relevant key insights into their revision. The authors have revised erroneous statements surrounding the radiative kernels as well as developed a more honest discussion surrounding the limitations and caveats of their methods and how they possibly influence their results, which were my two largest concerns. I am now comfortable in seeing this paper accepted and published.

Regarding the confusion surrounding the numbers in Smith et al.'s 2021 Table 7, it seems I had referred to the numbers in the pre-print version of the paper (Smith et al. 2020) and not the final published version, for which I apologize. I had not picked up on this discrepancy before.

Table 7, Smith et al. (2020), ACP Discussions

<https://doi.org/10.5194/acp-2019-1212>
Preprint. Discussion started: 20 January 2020
© Author(s) 2020. CC BY 4.0 License.

Table 7. As for table 3 but for 1850–2014 land-use forcing.

Model	ERF	IRF	Adj.	ts	ta_tr	ta_st	hus	cl
CanESM5	-0.08	-0.10	0.03	0.02	-0.02	0.01	-0.00	0.02
CESM2	-0.04	-0.08	0.05	-0.01	0.03	0.00	0.01	0.03
CNRM-ESM2-1	-0.07	-0.08	0.02	0.01	-0.02	-0.01	0.04	-0.00
GFDL-CM4	-0.33	-0.42	0.09	-0.04	0.09	0.00	-0.06	0.08
GISS-E2-1-G	-0.00	0.02	-0.02	-0.02	-0.02	0.01	0.02	-0.01
HadGEM3-GC31-LL	-0.11	-0.16	0.06	0.01	0.10	0.01	-0.05	-0.02
IPSL-CM6A-LR	-0.05	-0.11	0.07	-0.01	0.02	0.00	-0.01	0.07
MIROC6	-0.03	-0.10	0.08	-0.01	0.04	0.00	-0.04	0.10
MPI-ESM1-2-LR	-0.10	-0.01	-0.09	-0.01	0.01	0.01	-0.01	-0.10
MRI-ESM2-0	-0.17	-0.33	0.16	0.00	0.08	-0.00	-0.08	0.16
NorESM2-LM	0.26	-0.01	0.27	0.01	0.01	0.00	0.00	0.25
UKESM1-0-LL	-0.30	-0.28	-0.01	0.02	0.08	-0.01	-0.04	-0.06
Mean	-0.08	-0.14	0.06	-0.00	0.03	0.00	-0.02	0.04
St. dev.	0.14	0.13	0.09	0.02	0.04	0.01	0.03	0.09

Table 7, Smith et al. (2021), ACP

Table 7. As for Table 3 but for 1850–2014 land use forcing.

No.	Model	ERF	IRF	Adj.	ts	ta_tr	ta_st	hus	cl
2	CanESM5	-0.08	-0.10	0.03	0.02	-0.02	0.01	-0.00	0.02
3	CESM2	-0.04	-0.09	0.05	-0.01	0.03	0.00	0.01	0.04
5	CNRM-ESM2-1	-0.07	-0.09	0.03	0.01	-0.02	-0.01	0.04	0.00
6	EC-Earth3	-0.13	-0.13	-0.00	0.02	0.07	0.02	-0.03	-0.09
7	GFDL-CM4	-0.33	-0.41	0.08	-0.04	0.09	0.00	-0.06	0.08
8	GFDL-ESM4	-0.28	-0.27	-0.01	-0.03	0.08	0.01	-0.11	0.04
9	GISS-E2-1-G p1	-0.00	0.02	-0.02	-0.02	-0.02	0.01	0.02	-0.01
11	HadGEM3-GC31-LL	-0.11	-0.18	0.07	0.01	0.10	0.01	-0.05	-0.00
12	IPSL-CM6A-LR	-0.05	-0.09	0.05	-0.01	0.02	0.00	-0.01	0.05
13	MIROC6	-0.03	-0.07	0.04	-0.01	0.04	0.00	-0.04	0.04
14	MPI-ESM1-2-LR	-0.10	-0.06	-0.04	-0.01	0.01	0.01	-0.01	-0.05
15	MRI-ESM2-0	-0.17	-0.32	0.15	0.00	0.08	-0.00	-0.08	0.15
16	NorESM2-LM	0.26	-0.00	0.27	0.01	0.01	0.00	0.00	0.24
18	UKESM1-0-LL	-0.18	-0.18	0.00	0.00	0.04	0.01	-0.05	-0.00
	Mean	-0.09	-0.14	0.05	-0.00	0.04	0.01	-0.03	0.04
	SD	0.13	0.12	0.08	0.02	0.04	0.01	0.04	0.08

Dear Editors and Reviewer,

please find below our point-to-point responses to all Reviewer comments of revision 2 of COMMSENV-22-0963B

REVIEWER COMMENTS:

Reviewer #1 (Remarks to the Author):

I thank the authors for their efforts in addressing my biggest concerns. They demonstrated that they invested effort in becoming acquainted with the recommended literature and successfully extracted and integrated the relevant key insights into their revision. The authors have revised erroneous statements surrounding the radiative kernels as well as developed a more honest discussion surrounding the limitations and caveats of their methods and how they possibly influence their results, which were my two largest concerns. I am now comfortable in seeing this paper accepted and published.

Regarding the confusion surrounding the numbers in Smith et al.'s 2021 Table 7, it seems I had referred to the numbers in the pre-print version of the paper (Smith et al. 2020) and not the final published version, for which I apologize. I had not picked up on this discrepancy before.

Table 7, Smith et al. (2020), ACP Discussions

<https://doi.org/10.5194/acp-2019-1212>
Preprint. Discussion started: 20 January 2020
© Author(s) 2020. CC BY 4.0 License.

Table 7. As for table 3 but for 1850–2014 land-use forcing.

Model	ERF	IRF	Adj.	ts	ta_tr	ta_st	hus	cl
CanESM5	-0.08	-0.10	0.03	0.02	-0.02	0.01	-0.00	0.02
CESM2	-0.04	-0.08	0.05	-0.01	0.03	0.00	0.01	0.03
CNRM-ESM2-1	-0.07	-0.08	0.02	0.01	-0.02	-0.01	0.04	-0.00
GFDL-CM4	-0.33	-0.42	0.09	-0.04	0.09	0.00	-0.06	0.08
GISS-E2-1-G	-0.00	0.02	-0.02	-0.02	-0.02	0.01	0.02	-0.01
HadGEM3-GC31-LL	-0.11	-0.16	0.06	0.01	0.10	0.01	-0.05	-0.02
IPSL-CM6A-LR	-0.05	-0.11	0.07	-0.01	0.02	0.00	-0.01	0.07
MIROC6	-0.03	-0.10	0.08	-0.01	0.04	0.00	-0.04	0.10
MPI-ESM1-2-LR	-0.10	-0.01	-0.09	-0.01	0.01	0.01	-0.01	-0.10
MRI-ESM2-0	-0.17	-0.33	0.16	0.00	0.08	-0.00	-0.08	0.16
NorESM2-LM	0.26	-0.01	0.27	0.01	0.01	0.00	0.00	0.25
UKESM1-0-LL	-0.30	-0.28	-0.01	0.02	0.08	-0.01	-0.04	-0.06
Mean	-0.08	-0.14	0.06	-0.00	0.03	0.00	-0.02	0.04
St. dev.	0.14	0.13	0.09	0.02	0.04	0.01	0.03	0.09

Table 7, Smith et al. (2021), ACP

Table 7. As for Table 3 but for 1850–2014 land use forcing.

No.	Model	ERF	IRF	Adj.	ts	ta_tr	ta_st	hus	cl
2	CanESM5	-0.08	-0.10	0.03	0.02	-0.02	0.01	-0.00	0.02
3	CESM2	-0.04	-0.09	0.05	-0.01	0.03	0.00	0.01	0.04
5	CNRM-ESM2-1	-0.07	-0.09	0.03	0.01	-0.02	-0.01	0.04	0.00
6	EC-Earth3	-0.13	-0.13	-0.00	0.02	0.07	0.02	-0.03	-0.09
7	GFDL-CM4	-0.33	-0.41	0.08	-0.04	0.09	0.00	-0.06	0.08
8	GFDL-ESM4	-0.28	-0.27	-0.01	-0.03	0.08	0.01	-0.11	0.04
9	GISS-E2-1-G p1	-0.00	0.02	-0.02	-0.02	-0.02	0.01	0.02	-0.01
11	HadGEM3-GC31-LL	-0.11	-0.18	0.07	0.01	0.10	0.01	-0.05	-0.00
12	IPSL-CM6A-LR	-0.05	-0.09	0.05	-0.01	0.02	0.00	-0.01	0.05
13	MIROC6	-0.03	-0.07	0.04	-0.01	0.04	0.00	-0.04	0.04
14	MPI-ESM1-2-LR	-0.10	-0.06	-0.04	-0.01	0.01	0.01	-0.01	-0.05
15	MRI-ESM2-0	-0.17	-0.32	0.15	0.00	0.08	-0.00	-0.08	0.15
16	NorESM2-LM	0.26	-0.00	0.27	0.01	0.01	0.00	0.00	0.24
18	UKESM1-0-LL	-0.18	-0.18	0.00	0.00	0.04	0.01	-0.05	-0.00
	Mean	-0.09	-0.14	0.05	-0.00	0.04	0.01	-0.03	0.04
	SD	0.13	0.12	0.08	0.02	0.04	0.01	0.04	0.08

Thank you very much.